# WMVLM: Evaluating Diffusion Model Image Watermarking via Vision-Language Models

**Zijin Yang** [* 1 2] **Yu Sun** [* 3] **Kejiang Chen** [1 2] **Jiawei Zhao** [1 2] **Jun Jiang** [1 2] **Weiming Zhang** [1 2] **Nenghai Yu** [1 2]

## Abstract

Digital watermarking is essential for securing generated images from diffusion models. Accurate watermark evaluation is critical for algorithm development, yet existing methods have significant limitations: they lack a unified framework for both residual and semantic watermarks, provide results without interpretability, neglect comprehensive security considerations, and often use inappropriate metrics for semantic watermarks. To address these gaps, we propose **WMVLM**, the first unified and interpretable evaluation framework for diffusion model image **w**ater**m**arking via **v**ision-**l**anguage **m**odels (VLMs). We redefine quality and security metrics for each watermark type: residual watermarks are evaluated by artifact strength and erasure resistance, while semantic watermarks are assessed through latent distribution shifts. Moreover, we introduce a three-stage training strategy to progressively enable the model to achieve classification, scoring, and interpretable text generation. Experiments show WMVLM outperforms state-of-the-art VLMs with strong generalization across datasets, diffusion models, and watermarking methods.

## 1. Introduction

Diffusion models (Sohl-Dickstein et al., 2015; Ho et al., 2020; Song et al., 2020; Rombach et al., 2022) have achieved remarkable progress in image generation, but their potential misuse in creating misinformation or infringing copyrights demands effective watermarking solutions. Cur-

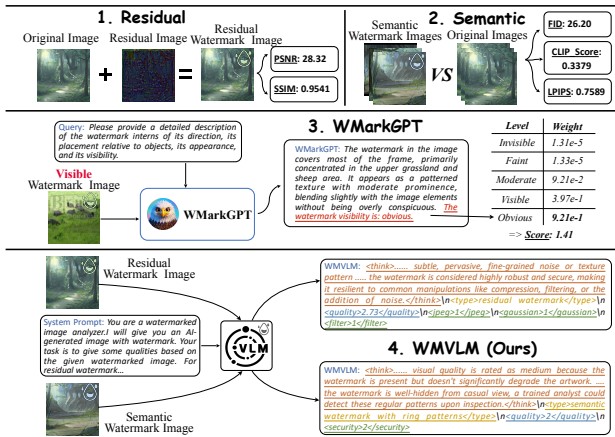

*Figure 1.* Comparison of evaluation paradigms. While traditional residual and semantic metrics provide only absolute numerical values, WMarkGPT introduces VLM-based interpretability for visible watermarks. Our WMVLM establishes a unified and interpretable framework specifically tailored to evaluate both visual quality and security in diffusion model image watermarking.

rent watermarking methods fall into two categories: (1) *Residual Watermarks* (Cox et al., 2007; Zhu et al., 2018; Zhang et al., 2019; Fernandez et al., 2023; Hu et al., 2024b; Lu et al., 2025), which embed information by modifying pixel-level or frequency-domain features but often introduces visible artifacts; (2) *Semantic Watermarks* (Wen et al., 2023; Yang et al., 2024; Ci et al., 2024; Gunn et al., 2025; Li et al., 2025a; Yang et al., 2025b;a), which alter latent representations to achieve high quality and imperceptibility.

Given the diversity of watermarking methods and the varied application requirements, systematic performance evaluation is essential. Existing evaluations focus on two key dimensions: (1) *Robustness*, assessing resistance to distortions such as image processing operations and erasure attacks (Zhao et al., 2024; An et al., 2024; Lin & Juarez, 2025); (2) *Visual Quality*, measuring the perceptual impact of water- mark embedding. For residual watermarks, quality is typically assessed from a pixel-level perspective using the Peak Signal-to-Noise Ratio (PSNR) (Gonzalez, 2009) and the Structural Similarity Index (SSIM) (Wang et al., 2004). In contrast, semantic watermarks are often evaluated using more semantically aware metrics, including

---
[*]Equal contribution [1]School of Cyber Science and Technology, University of Science and Technology of China, Anhui, China [2]Anhui Province Key Laboratory of Digital Security, Anhui, China [3]School of Computing, National University of Singapore, Singapore. Correspondence to: Kejiang Chen <chenkj@ustc.edu.cn>.

*Proceedings of the 43rd International Conference on Machine Learning*, Seoul, South Korea. PMLR 306, 2026. Copyright 2026 by the author(s).

the Fréchet Inception Distance (FID) (Heusel et al., 2017), CLIP Score (Radford et al., 2021), and Learned Perceptual Image Patch Similarity (LPIPS) (Zhang et al., 2018).

However, existing evaluation methods for diffusion model watermarking still exhibit significant limitations. First, residual and semantic watermarks rely on disparate metrics, lacking a unified evaluation framework for cross-paradigm comparison. Second, current metrics are predominantly numerical, providing quantitative results without interpretability to explain watermark effects on specific image attributes. Although WMarkGPT (Tan et al., 2025) recently introduced VLM-based (Touvron et al., 2023) semantic descriptions for improved interpretability, it focuses on visible watermarks and has not been applied to diffusion model watermarking. Moreover, WMarkGPT neglects security assessment, failing to evaluate robustness against malicious erasure attacks (Zhao et al., 2024; An et al., 2024; Lin & Juarez, 2025). Finally, traditional metrics for semantic watermarks are often inappropriate; they fail to distinguish between the inherent stochasticity of the diffusion process and the actual impact of watermark embedding.

These gaps collectively raise a central research question: **Can we design a unified and interpretable evaluation framework applicable to both residual and semantic watermarks, capable of assessing both visual quality and security through more appropriate metrics?**

We answer the aforementioned question affirmatively based on three key insights. First, recent advancements in VLMs demonstrate superior multimodal understanding and explanatory depth across diverse evaluation tasks (Xu et al.; Li et al., 2025b; Wu et al., 2025). By simultaneously processing low-level noise (relevant to residual watermarks) and high-level semantic content (relevant to semantic watermarks), VLMs provide a viable pathway toward a unified and interpretable framework. Second, an effective evaluation should differentiate security scenarios, where residual watermarks address typical online social networks (OSNs) interference (Wu et al., 2022) and semantic watermarks target sophisticated erasure attacks. Finally, since semantic watermarks manipulate latent representations, quality assessment should prioritize statistical analysis of latent distributions over image-space metrics, thereby mitigating the impact of inherent stochasticity on watermark evaluation.

Based on the above insights, we propose **WMVLM**, the first evaluation framework for diffusion model image **w**ater**m**arking via **v**ision-**l**anguage **m**odels. Our framework not only provides numerical scores across both visual quality and security dimensions, but also delivers interpretable textual outputs alongside the quantitative results.

We first redefine the scoring of visual quality and security. For residual watermarks, we associate visual qual-

ity with PSNR and evaluate security based on robustness against typical OSN distortions, including JPEG compression, Gaussian noise, and median filtering. For semantic watermarks, we perform large-scale sampling of latent representations and employ multiple hypothesis testing (Cramér, 1928; Von Mises, 1936; Jarque & Bera, 1987; D'agostino et al., 1990) to calculate the statistical significance ($p$-value) between watermarked and standard normal distributions $\mathcal{N}(0, I)$. This $p$-value quantifies the distributional deviation introduced by the watermark, where a higher value indicates that the latent distribution remains closer to $\mathcal{N}(0, I)$, thereby signifying superior visual quality and stronger security (Gunn et al., 2025; Yang et al., 2025b).

Upon defining these metrics, we proceed to the VLM training phase. Given that evaluating residual and semantic watermarks requires distinct response templates, training a VLM to simultaneously output scores and interpretable text is a significant challenge. We address this through a three-stage training process. The first stage, category and score pre-training, utilizes supervised fine-tuning (SFT) (Liu et al., 2023; Dai et al., 2023; Ouyang et al., 2022) to enable the VLM to classify watermark types and output scores based on corresponding templates. The second stage, explainability cold start, incorporates knowledge distillation from a powerful teacher model to guide the VLM in generating detailed textual explanations. To mitigate the pattern rigidity often associated with SFT (Chu et al., 2025), the third stage employs Group Relative Policy Optimization (GRPO) (Shao et al., 2024). This reinforcement learning phase uses a multi-faceted reward function to enhance generalization and stimulate the model's multimodal reasoning.

Extensive experiments on Stable Diffusion (Rombach et al., 2022) demonstrate that WMVLM outperforms SOTA open-source and closed-source VLMs. Furthermore, WMVLM exhibits strong generalization across datasets, diffusion models, and watermarking methods. Our contributions are summarized as follows:

- We propose WMVLM, the first unified and interpretable evaluation framework based on VLMs designed specifically for image watermarking in diffusion models. It effectively integrates both quantitative scoring and textual explanations, addressing a critical gap in existing evaluation approaches.

- We establish refined definitions of visual quality and security tailored to multi-paradigm evaluation, linking residual watermarks to PSNR and robustness against OSN-typical distortions, while assessing semantic watermarks via latent-space statistical significance through hypothesis testing.

- We introduce a three-stage training strategy that progressively trains the VLM through category and score

pre-training, explainability cold start, and generalization enhancement via GRPO. This structured approach enables stable learning of multi-paradigm evaluation while enhancing generalization and accuracy.

- We conduct comprehensive experiments on Stable Diffusion, demonstrating that WMVLM outperforms existing SOTA VLMs. WMVLM also exhibits strong generalization across datasets, diffusion models, and watermarking methods.

## 2. Related Work

### 2.1. Image Watermarking for Diffusion Models

Watermarking for diffusion models is mainly categorized into residual and semantic approaches. Residual watermarks embed patterns in the generated image. Post-processing methods (Cox et al., 2007; Zhang et al., 2019; Zhu et al., 2018; Fang et al., 2023; Hu et al., 2024a;b; Lu et al., 2025), applied after generation in the spatial or frequency domain without joint training, typically suffer from inferior visual quality and robustness. In contrast, generation-integrated methods fine-tune parts of the model (Fernandez et al., 2023; Zhao et al., 2023; Min et al., 2024; Wang et al., 2025) or insert dedicated modules (Feng et al., 2024; Fei et al., 2025) to embed the watermark during synthesis, improving performance yet often leaving detectable artifacts.

To move beyond residual patterns, Wen et al. (Wen et al., 2023) pioneered semantic watermarks via latent-space tree-ring patterns. Subsequent work diverged along two main trajectories: one exploring the capacity and traceability applications of the ring-based paradigm (Ci et al., 2024; Li et al., 2025a; Lee & Cho, 2025), and the other, initiated by Yang et al.'s distribution-preserving sampling approach (Yang et al., 2024), pursuing performance-lossless semantic watermarks. This latter direction has since addressed practical hurdles like key management (Gunn et al., 2025), parameter-mismatch robustness (Yang et al., 2025b), and diversity preservation (Yang et al., 2025a), steadily progressing toward real-world applicability.

### 2.2. Watermark Image Evaluation

The performance evaluation of image watermarks focuses on two core objectives: robustness and visual quality. Robustness measures the ability to withstand attacks while allowing successful extraction, with conventional testing covering common image-processing attacks (compression, noise, filtering). As watermarking shifts toward deep-network embedding, more advanced erasure attacks have emerged, i.e., Zhao et al. (Zhao et al., 2024) who proposed regeneration attacks using VAEs or diffusion models to remove non-semantically bound watermarks. While semantic watermarks generally resist such attacks, later stud-

ies (An et al., 2024; Lin & Juarez, 2025) have introduced adversarial-example-based erasure by leveraging gradients from a watermark discriminator.

Visual quality captures the perceptual impact of watermark embedding. Early post-processing watermarks, often leaving visible residuals, are evaluated with pixel-based metrics like PSNR (Gonzalez, 2009) and SSIM (Wang et al., 2004). For generation-integrated watermarking, perception-oriented metrics have been adopted, including FID (Heusel et al., 2017), CLIP-Score (Radford et al., 2021), and LPIPS (Zhang et al., 2018).

Nevertheless, these metrics remain limited: they are purely numerical, lack interpretability, and fail to explain how watermark embedding subjectively affects the image. To improve interpretability, Tan et al. (Tan et al., 2025) introduced WMarkGPT, which uses VLMs for explainable evaluation. However, WMarkGPT is incapable of evaluating advanced image watermarks in diffusion models and lacks consideration of security-related dimensions.

### 2.3. Vision-Language Model Training

Modern VLM training typically progresses from large-scale pre-training to post-training alignment via Supervised Fine-tuning (SFT) and Reinforcement Learning (RL).

**Supervised Fine-tuning.** SFT equips pre-trained models with instruction-following capabilities using curated image-text pairs (Liu et al., 2023; Dai et al., 2023). This bridges the modality gap, aligning visual features with linguistic semantics. However, SFT's reliance on maximizing the likelihood of static ground-truth tokens limits its ability to capture holistic, non-differentiable qualities and to explore solution paths beyond the training data (Ouyang et al., 2022).

**Reinforcement Learning Alignment.** RL addresses SFT's limitations by aligning models with complex rewards. While Proximal Policy Optimization (PPO) (Schulman et al., 2017) is widely adopted, its value network imposes significant memory overheads. Direct Preference Optimization (DPO) (Rafailov et al., 2023) offers a memory-efficient offline alternative but sacrifices on-policy exploration. Recently, Group Relative Policy Optimization (GRPO) (Shao et al., 2024) has emerged as an efficient solution. By replacing the value network with group-wise relative scoring, GRPO enables effective exploration without the computational burden of a critic.

## 3. Method

This section introduces WMVLM, the proposed evaluation framework for diffusion-model image watermarking. As illustrated in Fig.2, the process begins with VLM training preliminaries and the formal definition of quality and secu-

rity scores. We then present a three-stage training pipeline comprising category and score pre-training, an interpretability cold start, and generalization enhancement via GRPO. Refer to App. A for stage-specific prompts and App.B for evaluation examples.

### 3.1. Preliminaries

**Supervised Fine-tuning.** We employ Supervised Fine-tuning (SFT) to align the pre-trained VLM with the specific requirements of watermark evaluation. Given a dataset $\mathcal{D} = \{(Q_i, \mathbf{y}_i)\}_{i=1}^N$, where $Q_i$ denotes the multimodal input (encompassing the watermarked image $x_i$ and task instructions $q_i$) and $\mathbf{y}_i = (y_1, y_2, \ldots, y_N)$ represents the ground-truth response sequence, the training objective is to minimize the negative log-likelihood of the target tokens:

$$\mathcal{L}_{\text{SFT}}(\theta) = -\mathbb{E}_{(Q_i, \mathbf{y}_i) \sim \mathcal{D}} \left[ \sum_{t=1}^T \log \pi_\theta(y_t \mid Q_i, y_{<t}) \right], \quad (1)$$

where $\pi_\theta$ is the policy parameterized by $\theta$, and $y_{<t}$ denotes the context of preceding ground-truth tokens.

In our framework, SFT serves a dual purpose. First, it enables the model to effectively discern distinct watermark features, i.e., residual artifacts versus latent distribution shifts, and accurately map them to the corresponding score templates. Second, it provides a stable reference policy $\pi_{\text{ref}}$ that ensures well-formatted outputs, thereby acting as a cold start for the subsequent reinforcement learning stage.

**Group Relative Policy Optimization.** For each input $q$, GRPO samples a group of outputs $\{o^k\}_{k=1}^G$ from the old policy $\pi_{\theta_{\text{old}}}$. The objective function maximizes the expected reward while maintaining stability through a clipped probability ratio and a KL-divergence penalty:

$$\mathcal{J}_{\text{GRPO}}(\theta) = \mathbb{E}_{q \sim Q, \{o^k\}_{k=1}^G \sim \pi_{\theta_{\text{old}}}} \left[ \frac{1}{G} \sum_{k=1}^G \left( \mathcal{L}_{\text{clip}}^k(\theta) - \right. \right.$$
$$\left. \left. \beta D_{\text{KL}}(\pi_\theta(o^k|q) || \pi_{\text{ref}}(o^k|q)) \right) \right], \quad (2)$$

where $Q$ denotes the candidate question set, $\pi_{\text{ref}}$ denotes the reference SFT model, and $\beta$ controls the divergence penalty. The clipped surrogate term $\mathcal{L}_{\text{clip}}^k(\theta)$ is defined as:

$$\mathcal{L}_{\text{clip}}^k(\theta) = \min\left(\rho^k(\theta)A^k, \text{clip}(\rho^k(\theta), 1-\epsilon, 1+\epsilon)A^k\right), \quad (3)$$

where $\rho^k(\theta) = \frac{\pi_\theta(o^k|q)}{\pi_{\theta_{\text{old}}}(o^k|q)}$ represents the importance sampling ratio. To ensure training stability, it is clipped within $[1-\epsilon, 1+\epsilon]$.

Instead of using a learned value function to estimate the advantage, GRPO computes the advantage $A^k$ by normalizing the reward $r^k$ against the group statistics:

$$A^k = \frac{r^k - \mu_{\text{group}}}{\sigma_{\text{group}}}, \quad (4)$$

where the group mean $\mu_{\text{group}}$ and standard deviation $\sigma_{\text{group}}$ are calculated over the sampled outputs $\{r^k\}_{k=1}^G$ for the specific input $q$. This group-relative optimization encourages broader exploration by generating multiple responses per query and learning from their relative rankings, thereby strengthening generalization to complex, open-ended tasks.

### 3.2. Score Definition

To enable WMVLM to assess watermark quality and security, we establish distinct scoring criteria tailored to the characteristics of residual and semantic watermarks.

**Residual watermarks.** For quality assessment, we adopt PSNR, a classic metric for measuring distortion, and normalize its value into a floating-point score $v^{(\text{res})} \in [1, 5]$. For security evaluation, we focus on robustness against three classical online social network (OSN) attacks: JPEG compression, Gaussian noise, and median filtering. The security assessment is encoded as three binary labels $\mathbf{s}^{(\text{res})} = (s_j, s_g, s_f) \in \{0, 1\}^3$, indicating robustness against the corresponding attack types respectively.

**Semantic watermarks.** Semantic watermarks alter latent representation distributions to embed watermark messages. Following recent findings (An et al., 2024; Lin & Juarez, 2025) that distribution shifts not only lead to visual quality degradation but also serve as detectable signatures vulnerable to erasure attacks, we evaluate quality and security by measuring distributional deviation from standard normal distributions $\mathcal{N}(0, I)$. For each method, we generate watermarked latent representation samples and apply hypothesis tests (Cramér-von Mises test (Cramér, 1928; Von Mises, 1936), Jarque-Bera test (Jarque & Bera, 1987), and D'Agostino's K² test (D'agostino et al., 1990)) to compute $p$-value between the watermarked latent distribution and $\mathcal{N}(0, I)$. A higher $p$-value indicate closer alignment with $\mathcal{N}(0, I)$, corresponding to better quality and security. Both scores are categorized into three-level discrete scales: $v^{(\text{sem})} \in \{1, 2, 3\}$ and $s^{(\text{sem})} \in \{1, 2, 3\}$.

### 3.3. Category and Score Pre-training

VLMs have limited zero-shot capacity for detecting watermark features and require structured supervision to effectively learn categorization and scoring tasks. Therefore, in this initial phase, we train the model to correctly classify watermarks and output corresponding quality and security scores. Following the formal definition of these scoring metrics, the VLM is fine-tuned using the MLE loss as formulated in Eq.1, where the ground-truth supervision signals are $\mathbf{y}_{1i}^{(\text{res})} = (c_i^{(\text{res})}, v_i^{(\text{res})}, \mathbf{s}_i^{(\text{res})})$ for residual watermarks and $\mathbf{y}_{1i}^{(\text{sem})} = (c_i^{(\text{sem})}, v_i^{(\text{sem})}, s_i^{(\text{sem})})$ for semantic watermarks, with $c_i^{(\text{res})}$ and $c_i^{(\text{sem})}$ denoting the category labels.

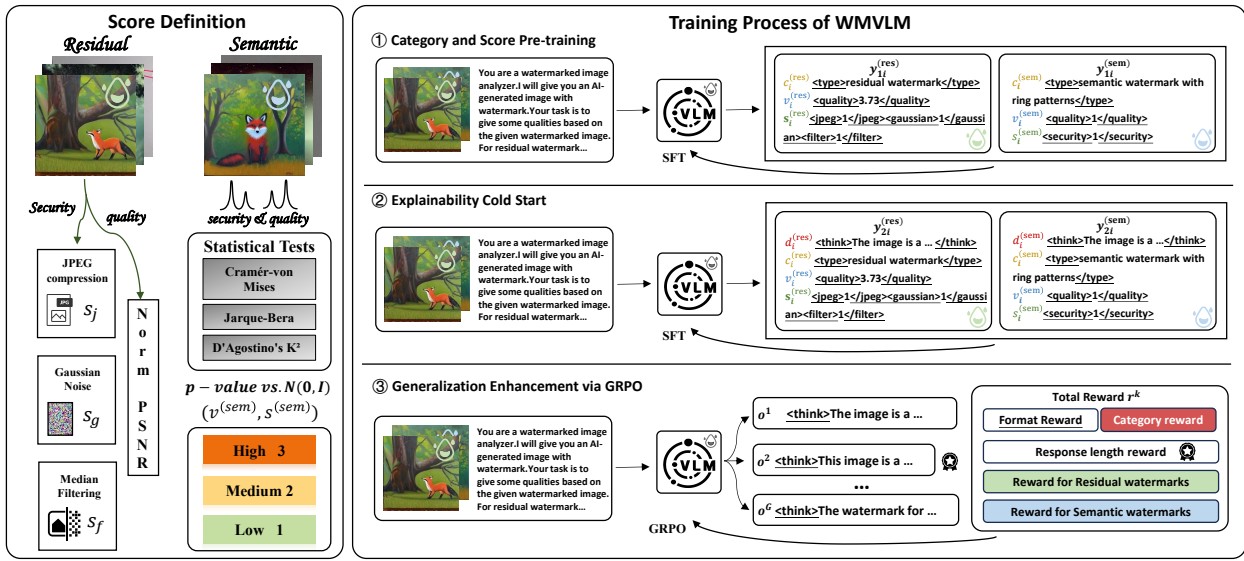

*Figure 2.* **WMVLM follows a three-stage training process**. After defining the quality and security scores, the model undergoes Category and Score Pre-training, Interpretability Cold Start, and Generalization Enhancement via GRPO.

## 3.4. Interpretability Cold Start

While the pre-trained VLM can classify watermark types and predict numerical scores, it lacks the ability to generate interpretable explanations. Directly applying GRPO at this stage risks training instability, as the model cannot reliably produce template-compliant responses. To address this issue, we introduce a cold start phase using knowledge distillation from a powerful teacher model. We employ Gemini-2.5-Pro (Comanici et al., 2025) to generate explanatory text for each watermarked image $x_i$ and its label $\mathbf{y}_{1i}$, where $\mathbf{y}_{1i} = \mathbf{y}_{1i}^{(\text{res})}$ for residual watermarks and $\mathbf{y}_{1i} = \mathbf{y}_{1i}^{(\text{sem})}$ for semantic watermarks. While the teacher model struggles to directly generate such explanations from visual input alone, it can effectively rationalize the given assessment results through post-hoc reasoning when provided with both $x_i$ and $\mathbf{y}_{1i}$. We prompt Gemini-2.5-Pro to analyze visual attributes, texture details, regional characteristics, watermark patterns, and their impact on quality and security (see App. A.1 for full prompt). Through this post-hoc reasoning process, the generated explanations $d_i^{(\text{res})}$ and $d_i^{(\text{sem})}$ are combined with the original labels to form the supervision signals $\mathbf{y}_{2i}^{(\text{res})} = (d_i^{(\text{res})}, c_i^{(\text{res})}, v_i^{(\text{res})}, \mathbf{s}_i^{(\text{res})})$ and $\mathbf{y}_{2i}^{(\text{sem})} = (d_i^{(\text{sem})}, c_i^{(\text{sem})}, v_i^{(\text{sem})}, s_i^{(\text{sem})})$ for this training stage.

## 3.5. Generalization Enhancement via GRPO

Since both preceding training stages rely on SFT, the resulting scores and text may exhibit pattern rigidity and limited adaptability (Chu et al., 2025). To enhance the generalization and diversity of the VLM's responses, we introduce GRPO in the third stage. For each watermarked image $x_i$, the policy model $\pi_\theta$ samples $G$ candidate responses

$\{o^k\}_{k=1}^G$, which are subsequently evaluated using a specifically designed reward function $R$ to assign relative rewards $\{r^k\}_{k=1}^G$. Concretely, both watermark types share the same format reward, category reward, and response length reward, while quality and security score rewards are tailored to the distinct characteristics of residual and semantic watermarks.

**Format reward** assesses whether the output adheres to the required structure. The response must contain interpretative text within <think></think> tags, watermark category within <type></type> tags, and quality score within <quality></quality> tags. For residual watermarks, security assessment comprises three components enclosed in <jpeg></jpeg>, <gaussian></gaussian>, and <filter></filter> tags, while semantic watermarks use a single <security></security> tag (see App.B for examples). Given the template complexity, we apply a strict penalty: $r_{\text{fmt}}^k = 0$ if and only if the format is fully correct ($fmt^k = fmt_{\text{gt}}^k$); otherwise, the final reward is set to $r^k = -10$ and no further reward components are computed.

**Category reward** evaluates whether the model correctly identifies the watermark type. If the prediction is correct, i.e., $c^k = c_{\text{gt}}^k$, the reward $r_{\text{cat}}^k$ is set to 1; otherwise, the final reward $r^k$ is set to 0, and no further reward components are computed.

**Response length reward** constrains the output length within a reasonable range, preventing extreme length deviations that could cause the model to drift from the cold start distribution and lead to training collapse. Given a preset target length $l_{\text{gt}}$ and the model output length $l^k$, the

reward is designed as below:

$$r_{\text{len}}^k = \begin{cases} 1 - \frac{|l^k - l_{\text{gt}}|}{\tau_{\text{len}}}, & \text{if } |l^k - l_{\text{gt}}| \le \tau_{\text{len}} \\ 0, & \text{otherwise} \end{cases}, \quad (5)$$

where $\tau_{\text{len}}$ is a length-tolerance threshold. When the absolute deviation between $l^k$ and $l_{\text{gt}}$ does not exceeds $\tau_{\text{len}}$, $r_{\text{len}}^k$ decreases linearly from 1 to 0. If the deviation exceeds $\tau_{\text{len}}$, $r_{\text{len}}^k$ is set to 0.

**Reward for Residual watermarks.** For residual watermarks, the quality score is a floating-point value. To encourage the predicted score $v^{k\,(\text{res})}$ to remain near the ground-truth value $v_{\text{gt}}^{k\,(\text{res})}$, we define a linear reward function:

$$r_{\text{qual}}^{k\,(\text{res})} = \begin{cases} 1 - \frac{|v^{k\,(\text{res})} - v_{\text{gt}}^{k\,(\text{res})}|}{\tau_{\text{qual}}}, & \text{if } |v^{k\,(\text{res})} - v_{\text{gt}}^{k\,(\text{res})}| \le \tau_{\text{qual}} \\ 0, & \text{otherwise} \end{cases}, \quad (6)$$

where the threshold $\tau_{\text{qual}}$ controls the allowable deviation between $v^{k\,(\text{res})}$ and $v_{\text{gt}}^{k\,(\text{res})}$.

For security assessment, which evaluates resilience against three noise types, sub-rewards $r_j^k$, $r_g^k$, and $r_f^k$ are set to 1 if the model correctly predicts the corresponding capability, and 0 otherwise. The total security reward is the average of these sub-rewards: $r_{\text{sec}}^{k\,(\text{res})} = (r_j^k + r_g^k + r_f^k)/3$.

**Reward for Semantic watermarks.** For semantic watermarks, both the quality and security scores are defined as discrete ordinal levels. The reward $r_{\text{qual}}^{k\,(\text{sem})}$ or $r_{\text{sec}}^{k\,(\text{sem})}$ is set to 1 if the prediction is correct, and 0 otherwise.

**Total Reward Design.** In summary, the reward integrates five dimensions: format compliance, category prediction, response length, quality assessment, and security evaluation. The final composite reward is formulated as follows:

$$r^k = \begin{cases} -10, & \text{if } fmt^k \ne fmt_{\text{gt}}^k \\ 0, & \text{if } fmt^k = fmt_{\text{gt}}^k \text{ and } c^k \ne c_{\text{gt}}^k \\ 1 + r_{\text{len}}^k + r_{\text{qual}}^k + r_{\text{sec}}^k, & \text{if } fmt^k = fmt_{\text{gt}}^k \text{ and } c^k = c_{\text{gt}}^k \end{cases} \quad (7)$$

where the calculations of $r_{\text{qual}}^k$ and $r_{\text{sec}}^k$ depend on the watermark type specified by $c_{\text{gt}}^k$, and are computed as defined in the corresponding reward formulations above.

## 4. Experiments

### 4.1. Experimental Setup

**Datasets.** Our dataset encompasses 9 representative watermarking methods: 6 residual watermarks including Dwt-Dct (Cox et al., 2007), RivaGAN (Zhang et al., 2019), HiD-DeN (Zhu et al., 2018), RobustWide (RW) (Hu et al., 2024b),

VINE (Lu et al., 2025), and Stable Signature (SS) (Fernandez et al., 2023), along with 3 semantic watermarks including RingID (Ci et al., 2024), Tree-Ring (Wen et al., 2023), and a performance-lossless (Lossless) category including Gaussian Shading (GS) (Yang et al., 2024), PRCW (Gunn et al., 2025), and Gaussian Shading++ (GS++) (Yang et al., 2025b). All images are generated using SD v2.1 (Rombach et al., 2022) for its broad compatibility with existing watermarking methods, with prompts from the Stable-Diffusion-Prompts-2.47M (FredZhang7, 2024), yielding 5,000 images for residual watermarks and 9,500 images for semantic watermarks. For each method, 500 images are reserved for testing, with the rest for training. During the cold start, we generate 500 interpretable explanations per method using Gemini-2.5-Pro (Comanici et al., 2025). See Appendix D for detailed data splits and definitions of evaluation scores.

**Baselines.** To comprehensively evaluate the effectiveness of our proposed WMVLM, we compare it against a set of SOTA VLMs serving as baselines. This includes 3 closed-source models: GPT-5 (OpenAI, 2025), Claude-Opus-4.5 (Anthropic, 2025), and Gemini-3-Pro (Google, 2025); as well as 8 open-source models: LLaVA-v1.6 (7B, 13B) (Liu et al., 2024), Qwen3-VL (4B, 8B, 32B) (Team, 2025b), and Gemma-3 (4B, 12B, 27B) (Team, 2025a). All baseline models are evaluated under a zero-shot setting.

**Evaluation Metrics.** We evaluate continuous quality scores for residual watermarks using Pearson Linear Correlation Coefficient (PLCC) (Sedgwick, 2012) and the Spearman Rank-order Correlation Coefficient (SRCC) (Sedgwick, 2014). For discrete metrics, such as residual security and all semantic evaluation scores, we employ prediction accuracy as the primary assessment criterion.

**Implement Details.** WMVLM utilizes Qwen3-VL-8B-Instruct (Team, 2025b) as the backbone. Category and score pre-training uses a $10^{-5}$ learning rate over 6 epochs with a batch size of 128. The cold start stage employs a $10^{-6}$ learning rate for 8 epochs with a batch size of 64. For generalization enhancement, we train for 3 epochs with a batch size of 8 and a $10^{-6}$ learning rate, configuring GRPO with group size $G = 8$, divergence penalty $\beta = 0.1$, target response length $l_{\text{gt}} = 850$, length tolerance $\tau_{\text{len}} = 50$, and quality tolerance $\tau_{\text{qual}} = 0.3$. Training completes in approximately 24 hours on 8 NVIDIA RTX PRO 6000 GPUs.

### 4.2. Main Results

**Overall Comparison.** We first conduct a comprehensive comparison between WMVLM and the baseline models, with the experimental results shown in Tab.1. For residual watermarks, WMVLM achieves PLCC and SRCC values both above 0.7 for quality score evaluation, and its accuracy for security score assessment reaches approximately 99%. For semantic watermarks, the accuracy for both quality and

*Table 1.* Overall Comparison. Performance is reported as PLCC / SRCC / Average security accuracy (three noise types) for residual watermarks, and as Quality accuracy / Security accuracy for semantic watermarks.

| Models | Residual Watermarks | | | | | | Semantic Watermarks | | |
|---|---|---|---|---|---|---|---|---|---|
| | DwtDct | RivaGAN | HiDDeN | RW | VINE | SS | RingID | Tree-Ring | Lossless |
| GPT-5 | 0.064/0.013/0.000 | -0.070/-0.025/0.514 | 0.008/0.037/0.722 | 0.211/0.206/0.386 | 0.067/0.052/0.500 | 0.163/0.173/0.796 | 0.214/0.214 | 0.006/0.006 | 0.558/0.558 |
| Claude-Opus-4.5 | 0.067/0.044/0.002 | -0.139/-0.008/0.142 | 0.079/0.067/0.300 | -0.016/-0.025/0.106 | 0.219/0.194/0.182 | 0.272/0.281/0.490 | 0.042/0.138 | 0.324/0.270 | 0.572/0.576 |
| Gemini-3-Pro | 0.090/0.028/0.002 | 0.043/0.092/0.419 | 0.159/0.123/0.784 | 0.188/0.152/0.316 | 0.071/0.062/0.736 | 0.170/0.140/0.828 | 0.054/0.060 | 0.012/0.008 | 0.716/0.716 |
| LLaVA-v1.6-7B | 0.376/0.400/0.000 | 0.127/-0.224/0.000 | -0.098/-0.316/0.000 | 0.741/0.617/0.000 | -0.033/0.577/0.000 | 0.106/0.436/0.000 | 0.000/0.002 | 0.006/0.008 | 0.008/0.008 |
| LLaVA-v1.6-13B | -0.066/-0.086/0.000 | -0.035/-0.007/0.008 | -0.225/-0.243/0.009 | -0.093/-0.316/0.000 | 0.009/0.017/0.015 | 0.109/0.145/0.003 | 0.000/0.002 | 0.000/0.002 | 0.082/0.082 |
| Qwen-VL-4B | -0.121/0.002/0.006 | 0.442/0.545/0.020 | -0.163/-0.259/0.030 | 0.703/0.232/0.011 | 0.420/0.326/0.020 | 0.166/-0.010/0.081 | 0.014/0.022 | 0.002/0.002 | 0.890/0.890 |
| Qwen3-VL-8B | 0.205/0.500/0.002 | -0.146/0.004/0.010 | 0.146/0.169/0.016 | 0.953/0.961/0.004 | -0.311/-0.240/0.014 | -0.167/-0.299/0.048 | 0.046/0.054 | 0.008/0.002 | 0.908/0.908 |
| Qwen3-VL-32B | -0.135/-0.137/0.000 | 0.013/-0.094/0.044 | -0.042/-0.063/0.154 | 0.278/0.307/0.034 | -0.064/-0.056/0.065 | -0.222/-0.210/0.217 | 0.046/0.054 | 0.020/0.108 | 0.844/0.830 |
| Gemma-3-4B | 0.042/-0.076/0.013 | -0.167/-0.140/0.047 | 0.206/0.237/0.315 | 0.158/0.127/0.245 | -0.046/-0.031/0.296 | 0.259/0.275/0.316 | 0.006/0.006 | 0.246/0.246 | 0.078/0.140 |
| Gemma-3-12B | 0.245/0.111/0.043 | -0.193/-0.155/0.153 | 0.117/0.038/0.205 | 0.311/0.269/0.111 | 0.230/0.187/0.125 | 0.262/0.253/0.275 | 0.068/0.074 | 0.712/0.706 | 0.086/0.088 |
| Gemma-3-27B | 0.153/0.088/0.036 | -0.167/-0.142/0.399 | 0.064/0.017/0.485 | -0.010/-0.068/0.352 | 0.032/0.047/0.369 | 0.223/0.224/0.565 | 0.042/0.064 | 0.414/0.410 | 0.018/0.014 |
| **WMVLM** | **0.952/0.957/0.988** | **0.885/0.875/0.990** | **0.712/0.708/0.998** | **0.961/0.973/0.994** | **0.983/0.986/1.000** | **0.981/0.985/0.998** | **0.852/0.852** | **0.956/0.956** | **0.910/0.910** |

*Table 2.* Cross-dataset Results. Performance is reported as PLCC / SRCC / Average security accuracy (three noise types) for residual watermarks, and as Quality accuracy / Security accuracy for semantic watermarks.

| Models | Residual Watermarks | | | | | | Semantic Watermarks | | |
|---|---|---|---|---|---|---|---|---|---|
| | DwtDct | RivaGAN | HiDDeN | RW | VINE | SS | RingID | Tree-Ring | Lossless |
| GPT-5 | -0.089/-0.065/0.000 | 0.000/0.056/0.834 | 0.081/0.058/0.921 | 0.011/0.006/0.723 | -0.040/-0.052 | 0.672/0.002/0.952 | 0.038/0.038 | 0.000/0.000 | 0.424/0.424 |
| Claude-Opus-4.5 | -0.060/0.044/0.109 | -0.009/-0.050/0.173 | 0.060/0.038/0.252 | -0.072/-0.101/0.105 | 0.201/0.224/0.175 | 0.351/0.299/0.435 | 0.002/0.026 | 0.226/0.208 | 0.584/0.588 |
| Gemini-3-Pro | 0.007/-0.017/0.002 | -0.022/-0.063/0.434 | 0.001/-0.002/0.882 | 0.030/-0.007/0.318 | 0.016/0.035/0.662 | 0.162/0.144/0.888 | 0.026/0.026 | 0.006/0.006 | 0.790/0.790 |
| LLaVA-v1.6-7B | -0.255/-0.605/0.000 | -0.277/-0.088/0.000 | 0.265/0.328/0.000 | 0.568/0.444/0.000 | 0.356/0.435/0.000 | 0.161/0.006/0.000 | 0.000/0.004 | 0.000/0.008 | 0.014/0.016 |
| LLaVA-v1.6-13B | -0.316/-0.244/0.000 | 0.138/0.236/0.012 | 0.033/0.015/0.006 | 0.116/0.123/0.009 | -0.018/-0.038/0.007 | -0.021/-0.047/0.014 | 0.000/0.000 | 0.000/0.006 | 0.180/0.176 |
| Qwen3-VL-4B | -0.305/-0.200/0.002 | 0.392/0.415/0.002 | 0.245/0.261/0.026 | -0.980/-0.992/0.006 | 0.157/0.215/0.037 | 0.125/0.184/0.093 | 0.004/0.004 | 0.000/0.000 | 0.956/0.956 |
| Qwen3-VL-8B | 0.198/0.421/0.004 | 0.032/0.058/0.008 | -0.026/0.055/0.019 | 0.910/0.933/0.002 | 0.475/0.168/0.015 | -0.491/-0.343/0.065 | 0.010/0.010 | 0.000/0.000 | **0.992/0.992** |
| Qwen3-VL-32B | 0.114/0.062/0.000 | -0.307/-0.447/0.026 | 0.136/0.171/0.246 | 0.154/0.124/0.030 | -0.019/-0.016/0.081 | 0.002/0.009/0.350 | 0.012/0.026 | 0.008/0.004 | 0.912/0.910 |
| Gemma-3-4B | 0.043/0.007/0.018 | 0.060/-0.100/0.213 | -0.085/-0.130/0.313 | -0.098/-0.092/0.205 | 0.185/0.171/0.244 | 0.095/0.091/0.343 | 0.000/0.000 | 0.078/0.078 | 0.336/0.406 |
| Gemma-3-12B | 0.111/0.037/0.041 | -0.140/-0.080/0.128 | -0.042/0.031/0.204 | -0.018/-0.061/0.072 | 0.092/0.038/0.094 | 0.211/0.221/0.317 | 0.020/0.020 | 0.658/0.658 | 0.206/0.208 |
| Gemma-3-27B | 0.083/0.056/0.215 | -0.016/-0.0204/0.432 | 0.055/0.083/0.566 | 0.080/0.036/0.424 | 0.127/0.112/0.397 | 0.195/0.183/0.588 | 0.020/0.022 | 0.340/0.338 | 0.024/0.024 |
| **WMVLM** | **0.933/0.922/0.994** | **0.781/0.803/0.994** | **0.641/0.625/0.990** | **0.964/0.971/0.982** | **0.977/0.977/0.995** | **0.952/0.964/0.984** | **0.746/0.746** | **0.954/0.954** | 0.904/0.904 |

security evaluation is comparatively lower, which can be attributed to the more subtle and latent characteristics of semantic watermarks. Nonetheless, the accuracy remains above 85%, which is still highly competitive.

Among baseline methods, closed-source models generally outperform open-source counterparts. Notably, GPT-5 and Gemini-3-Pro can reliably detect the watermark presence in methods such as HiDDeN and Stable Signature, yet they fall short in predicting quality scores and evaluating semantic watermarks. It is worth mentioning that while the Qwen3 series can correctly identify performance-lossless watermarks, it tends to only output predictions for this specific category, limiting its general applicability. Overall, WMVLM significantly surpasses all baselines, providing a unified, interpretable, and accurate framework for assessing both the quality and security of diverse watermarking techniques.

**Cross-dataset Results.** To verify the cross-dataset generalization ability of WMVLM, we generate 500 watermarked images per method using 500 randomly selected prompts from the MS-COCO Validation Set (Lin et al., 2014). The experimental results presented in Tab.2 show that the accuracy of RingID decreases by approximately 10%. Nevertheless, the proposed WMVLM achieves performance comparable to that reported in Tab.1, while still outperforming all baseline methods. This demonstrates its robust generalization capability across datasets.

**Cross-model Results.** To evaluate cross-model general-

ization, we generate 500 watermarked images per method using SD v1.4 (Rombach et al., 2022) with the original test prompts. Results in Tab.3, indicate a clear performance drop for the Stable Signature, which embeds watermarks directly within the generation process, with the PLCC decreasing by approximately 0.6. This demonstrates that WMVLM is sensitive to architectural changes in diffusion models when watermarks are deeply integrated into its pipeline. Nevertheless, our method outperforms all baseline methods, confirming its excellent cross-model generalization ability.

**Cross-method Results.** To evaluate cross-method generalization, we generate 500 watermarked images for each of the following approaches: GaussMarker (Li et al., 2025a), T2SMark (Yang et al., 2025a), and original generated images (ori), which serve as a performance-lossless watermarking. The experimental results in Tab.4 show that all prediction accuracies exceed 87%. Furthermore, WMVLM surpasses all baseline methods in overall performance, demonstrating strong generalization across diverse methods.

### 4.3. Ablation Studies

In this section, we conduct ablation studies on the training strategy to validate the necessity of the proposed three-stage pipeline. Here, $S_1$, $S_2$, and $S_3$ denote the category and score pre-training, the explainability cold start, and the generalization enhancement via GRPO, respectively.

*Table 3.* Cross-model Results. Performance is reported as PLCC / SRCC / Average security accuracy (three noise types) for residual watermarks, and as Quality accuracy / Security accuracy for semantic watermarks.

| Models | Residual Watermarks | | | | | | Semantic Watermarks | | |
|---|---|---|---|---|---|---|---|---|---|
| | DwtDct | RivaGAN | HiDDeN | RW | VINE | SS | RingID | Tree-Ring | Lossless |
| GPT-5 | 0.120/0.106/0.000 | -0.116/-0.136/0.637 | 0.013/0.041/0.836 | 0.176/0.168/0.528 | 0.065/0.053/0.628 | 0.171/0.132/0.584 | 0.102/0.106 | 0.010/0.006 | 0.432/0.430 |
| Claude-Opus-4.5 | 0.282/0.005/0.092 | -0.203/-0.212/0.117 | 0.270/0.272/0.163 | 0.257/0.259/0.107 | 0.124/0.142/0.160 | 0.251/0.120/0.144 | 0.038/0.098 | 0.430/0.394 | 0.376/0.378 |
| Gemini-3-Pro | 0.257/0.216/0.000 | -0.032/-0.048/0.432 | 0.102/0.087/0.764 | 0.186/0.147/0.355 | 0.152/0.165/0.750 | 0.168/0.155/0.501 | 0.030/0.036 | 0.024/0.036 | 0.622/0.622 |
| LLaVA-v1.6-7B | 0.298/0.457/0.000 | -0.386/-0.141/0.024 | -0.002/0.010/0.000 | -0.589/-0.395/0.000 | 0.861/0.949/0.000 | -0.010/0.161/0.000 | 0.000/0.008 | 0.010/0.012 | 0.004/0.006 |
| LLaVA-v1.6-13B | 0.003/0.001/0.000 | -0.151/-0.200/0.009 | 0.004/-0.003/0.011 | -0.125/-0.087/0.008 | 0.166/0.101/0.014 | -0.032/0.018/0.009 | 0.000/0.000 | 0.000/0.000 | 0.090/0.090 |
| Qwen3-VL-4B | -0.432/0.218/0.000 | 0.375/0.577/0.010 | 0.237/0.132/0.019 | 0.051/-0.055/0.013 | -0.064/-0.200/0.024 | -0.188/-0.335/0.008 | 0.018/0.026 | 0.000/0.002 | 0.926/0.924 |
| Qwen3-VL-8B | 0.921/0.500/0.000 | -0.273/-0.500/0.006 | 0.318/0.409/0.022 | 0.575/0.556/0.013 | -0.846/-0.400/0.008 | -0.922/-0.987/0.006 | 0.058/0.062 | 0.004/0.002 | **0.932/0.934** |
| Qwen3-VL-32B | 0.337/0.387/0.000 | -0.106/-0.072/0.052 | -0.023/-0.032/0.110 | -0.290/-0.256/0.047 | 0.352/0.291/0.077 | 0.185/0.175/0.053 | 0.062/0.076 | 0.022/0.008 | 0.852/0.844 |
| Gemma-3-4B | 0.187/-0.076/0.022 | -0.131/-0.092/0.331 | -0.023/-0.083/0.328 | 0.151/0.136/0.284 | 0.239/0.221/0.315 | 0.151/0.129/0.283 | 0.004/0.004 | 0.276/0.276 | 0.076/0.130 |
| Gemma-3-12B | 0.002/-0.005/0.040 | -0.184/-0.105/0.165 | 0.115/0.184/0.174 | 0.074/0.081/0.127 | 0.105/0.124/0.117 | 0.137/0.113/0.106 | 0.056/0.062 | 0.648/0.636 | 0.102/0.102 |
| Gemma-3-27B | 0.123/0.058/0.175 | -0.248/-0.170/0.420 | 0.037/0.010/0.519 | 0.053/0.025/0.405 | 0.145/0.142/0.423 | 0.098/0.017/0.439 | 0.068/0.078 | 0.368/0.358 | 0.004/0.004 |
| **WMVLM** | **0.951/0.945/0.976** | **0.810/0.800/0.992** | **0.656/0.612/0.998** | **0.849/0.952/0.990** | **0.978/0.982/0.992** | **0.368/0.725/0.838** | **0.792/0.792** | **0.890/0.890** | 0.856/0.856 |

*Table 5.* Ablation study on training strategy of WMVLM. Performance is reported as PLCC / SRCC / Average security accuracy (three noise types) for residual watermarks, and as Quality accuracy / Security accuracy for semantic watermarks. **Bold** represents the best, underline represents the second best.

| Training Stage | | | Residual Watermarks | | | | | | Semantic Watermarks | | |
|---|---|---|---|---|---|---|---|---|---|---|---|
| $S_1$ | $S_2$ | $S_3$ | DwtDct | RivaGAN | HiDDeN | RW | VINE | SS | RingID | Tree-Ring | Lossless |
| | ✓ | | 0.048/0.075/0.106 | -0.019/-0.057/0.616 | -0.057/-0.036/0.870 | 0.058/0.038/0.556 | -0.008/-0.044/0.704 | 0.148/0.148/0.910 | 0.106/0.106 | 0.216/0.216 | 0.154/0.154 |
| | ✓ | ✓ | -0.036/0.046/0.000 | 0.048/0.071/0.788 | 0.023/0.004/0.740 | -0.015/0.018/0.796 | -0.103/-0.106/0.730 | -0.050/-0.086/0.775 | 0.000/0.000 | 0.000/0.002 | 0.000/0.000 |
| | | ✓ | 0.412/0.332/0.000 | 0.819/0.802/0.004 | 0.553/0.492/0.016 | 0.189/0.105/0.006 | 0.369/0.214/0.012 | -0.156/-0.018/0.051 | 0.056/0.060 | 0.004/0.000 | 0.916/0.914 |
| ✓ | | | **0.961/0.967/0.994** | 0.195/**0.882**/0.990 | 0.486/0.617/0.996 | **0.967**/0.985/0.998 | **0.984/0.987/1.000** | **0.990/0.990/1.000** | **0.922/0.922** | 0.950/0.950 | **0.946/0.946** |
| ✓ | ✓ | | 0.931/0.936/0.992 | 0.185/0.853/0.982 | 0.591/0.640/**1.000** | 0.869/0.970/0.992 | 0.950/0.978/0.994 | 0.979/0.985/**1.000** | 0.848/0.848 | 0.944/0.944 | 0.844/0.844 |
| ✓ | ✓ | ✓ | 0.952/0.957/0.988 | **0.885**/0.875/**0.990** | **0.712/0.708**/0.998 | 0.961/0.973/**0.994** | 0.983/0.986/**1.000** | 0.981/**0.985**/0.998 | 0.852/0.852 | **0.956/0.956** | 0.910/0.910 |

*Table 4.* Cross-method Results. Performance is reported as Quality accuracy / Security accuracy.

| Models | GaussMarker | T2SMark | Ori |
|---|---|---|---|
| GPT-5 | 0.004/0.000 | 0.676/0.674 | 0.612/0.610 |
| Claude-Opus-4.5 | 0.320/0.300 | 0.556/0.556 | 0.536/0.536 |
| Gemini-3-Pro | 0.010/0.004 | 0.700/0.700 | 0.704/0.706 |
| LLaVA-v1.6-7B | 0.000/0.012 | 0.008/0.010 | 0.012/0.014 |
| LLaVA-v1.6-13B | 0.000/0.000 | 0.106/0.106 | 0.084/0.084 |
| Qwen3-VL-4B | 0.000/0.000 | 0.890/0.894 | 0.874/0.874 |
| Qwen3-VL-8B | 0.002/0.000 | 0.902/0.910 | **0.900/0.902** |
| Qwen3-VL-32B | 0.012/0.008 | 0.906/0.890 | 0.844/0.836 |
| Gemma-3-4B | 0.254/0.254 | 0.102/0.166 | 0.120/0.178 |
| Gemma-3-12B | 0.684/0.678 | 0.096/0.096 | 0.100/0.098 |
| Gemma-3-27B | 0.354/0.342 | 0.002/0.002 | 0.012/0.012 |
| **WMVLM** | **0.956/0.956** | **0.930/0.930** | 0.874/0.874 |

inability to recognize semantic watermarks and leads to unsatisfactory performance, even after exploration via GRPO.

The third row of Tab.5 further illustrates the effect of omitting both $S_1$ and $S_2$. Without $S_1$, the model fails to acquire meaningful watermark features. Moreover, the absence of $S_2$ leaves the model without format guidance, causing it to frequently incur format errors during GRPO and thus receive low rewards, which severely hampers effective exploration. Consequently, the model exhibits behavior similar to that of an untrained model consistently outputting only the "performance-lossless watermark" category.

In summary, our three-stage training strategy is effective, with each stage being indispensable to the final performance. Additional ablation studies on hyperparameters sensitivity are provided in the App.E.

The last three rows of Tab.5 indicate that $S_2$ degrades the performance achieved after $S_1$, evidenced by lower PLCC for residual methods like RobustWide and VINE, and reduced accuracy for semantic methods like RingID and performance-lossless watermarking methods. However, subsequent $S_3$ yields further improvements, especially in the PLCC and SRCC for RivaGAN and HiDDeN. This suggests that exploration-driven GRPO better captures the link between watermark characteristics and quality, effectively mitigating the pattern rigidity from preceding SFT phases.

The first two rows of Tab.5 show that omitting $S_1$ hinders the model from learning discriminative watermark features. Relying solely on the limited data from $S_2$ results in an

## 5. Conclusion, Limitation and Future Work

In this paper, we propose WMVLM for evaluating image watermarking in diffusion models. We first define quality and security scores for both residual and semantic watermarks. A three-stage training strategy is then introduced to enable the model to accurately identify different watermarking paradigms, produce precise scores, and generate interpretable textual descriptions. Experimental results show that WMVLM outperforms SOTA VLMs and exhibits strong generalization capability.

While WMVLM achieves promising performance with GRPO, it remains dependent on SFT and thus requires

substantial data preparation. Future work could leverage more advanced reinforcement learning algorithms to enable accurate watermark evaluation without relying on SFT. Furthermore, WMVLM focuses on image watermarking in diffusion models; extending the evaluation framework to other modalities represents a valuable direction.

## Acknowledgments

This work was supported by National Natural Science Foundation of China (Grants U2336206, U2436601 and 62472398) and by the New Generation Artificial Intelligence-National Science and Technology Major Project (No. 2025ZD0123203).

## Impact Statement

Digital watermarking ensures the content security of diffusion-generated images, making reliable evaluation frameworks essential for technical advancement. WMVLM introduces a unified and interpretable evaluation framework specifically for this domain. We believe that WMVLM can effectively evaluate mainstream image watermarking algorithms for diffusion models, offering critical guidance to steer the design and advancement of future techniques in this field.

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

# A. Prompt Design

## A.1. Prompt for Knowledge Distillation

This subsection presents the system prompt used in the Interpretability Cold Start stage for knowledge Distillation, which is input to Gemini-2.5-Pro (Comanici et al., 2025) to obtain interpretable textual descriptions. The prompt instructs Gemini-2.5-Pro to analyze visual attributes, texture details, regional characteristics, watermark patterns, and their impact on quality and security.

---

**System Prompt**

You are a watermarked image analyzer.

I will give you an AI-generated image with a watermark and some of its qualities.

Your task is to give a thinking process based on the given watermarked image and qualities.

You must carefully examine the visual attributes, texture details, and regional characteristics of the image to identify potential watermark features, infer watermark patterns, and assess how they affect image quality and security.

Your response should be as detailed as possible, but the specific wording and sentence structure are not limited to the examples I have provided.

Your answer should be enclosed in <think>...</think>.

Attention: You should pretend that you are analyzing the watermarked image without seeing its qualities, but your final conclusion should be based on the given scores.

For residual watermark, E.g.:
User:
<some image>
<type>residual watermark</type>
<quality>2.72</quality>(a floating-point number between 1.0 and 5.0)
<jpeg>1</jpeg>(robustness indicators (each 0 or 1))
<gaussian>1</gaussian>(robustness indicators (each 0 or 1))
<filter>1</filter>(robustness indicators (each 0 or 1))
Assistant:
<think>The image depicts a traditional East Asian temple, rendered in a highly stylized, almost painterly manner. Upon initial observation, the entire image appears to be affected by a strong artistic filter. However, a closer look reveals that this effect is not uniform and results in significant visual artifacts and degradation of detail. The textures across the image, from the stone steps in the foreground to the roof tiles and the ornate gate, are blotchy and smeared. Edges are ill-defined, and fine details that one would expect in an architectural photograph are almost entirely absent, replaced by coarse, noisy patterns. For instance, the columns of the main gate are not smooth but are covered in a distorted, swirling texture. The sky in the upper left also shows unnatural blocky and smeared patterns rather than a smooth gradient or cloud texture. These characteristics are highly indicative of a strong visible watermark. The watermarking algorithm seems to have aggressively manipulated the image data, leading to these widespread artifacts. Given the severe impact on visual quality, it is logical to conclude that the watermark is embedded with great strength. This high strength is a trade-off for robustness, suggesting that the watermark is designed to be resilient against various common image manipulations such as compression, noise addition, and filtering. Therefore, it should be able to withstand common distortions found on social media platforms.</think>

For watermark-free or performance-lossless semantic watermark, E.g.:
User:
<some image>
<type>watermark-free or performance-lossless semantic watermark</type>
<quality>3</quality>(an integer from 1 to 3, 1 = low, 3 = high)
<security>3</security>(an integer from 1 to 3, 1 = low, 3 = high)
Assistant:
<think> The image presents a highly detailed and intricate circular design, resembling an abstract medallion or a complex compass. The overall visual impression is one of high quality, with sharp lines, vibrant colors, and

---

clear details throughout the complex patterns. Upon close examination, there are no visible artifacts such as blockiness, color banding, or distortions that would typically suggest the presence of a visible watermark. The image appears smooth and coherent. Furthermore, I am looking for evidence of semantic watermarks, such as ring patterns. The subject of the image is itself a series of concentric circles and complex radial patterns. This inherent circular geometry makes it exceedingly difficult, if not impossible, to visually identify any superimposed, artificial ring structures that might constitute a watermark. Any such pattern would be perfectly camouflaged within the intended design of the object. Given the complete absence of any perceivable visual degradation or tell-tale watermarking artifacts, the image is most likely either watermark-free or embedded with a performance-lossless semantic watermark. This type of watermark is integrated into the image generation process without leaving any negative impact on visual quality. Consequently, the quality is high. Because there are no detectable patterns for an adversary to analyze or attack, the watermark's security is also very strong.</think>

For semantic watermark with ring patterns, E.g.:
User:
<some image>
<type>semantic watermark with ring patterns</type>
<quality>1</quality>(an integer from 1 to 3, 1 = low, 3 = high)
<security>1</security>(an integer from 1 to 3, 1 = low, 3 = high)
Assistant:
<think>The image presented is a highly detailed, black-and-white portrait of a man with an intense, angry expression, rendered in a style reminiscent of pencil sketching or comic book inking. Upon a more detailed examination of the textures and shading, particularly in the smoother areas like the forehead, cheeks, and the large plates of armor on the chest, one might notice that the shading patterns are not entirely organic. The lines and textures seem to follow subtle, large-radius concentric curves. These circular patterns are not immediately obvious because they are skillfully blended with the hatching and shading of the art style itself. This suggests the presence of an embedded pattern rather than a simple overlay or visible effect. This type of subtle, integrated pattern is characteristic of a semantic watermark, specifically one that uses ring patterns. The watermark doesn't significantly degrade the image's aesthetic, which aligns with a medium quality designation. The details remain clear, but the underlying artificial pattern is detectable upon close scrutiny. The security is also considered medium; while a casual observer would likely miss the watermark entirely, a trained analyst looking for frequency-domain artifacts or specific geometric patterns could potentially detect and analyze these rings. The watermark is well-hidden but not completely undetectable.</think>

## A.2. Prompts for Category and Score Pre-training

This subsection presents the system prompt and user prompt used in the Category and Score Pre-training stage. The prompts instruct the model to first determine the watermark category and then output the corresponding quality and security scores based on the identified watermark type.

---
System Prompt

You are a watermarked image analyzer.
I will give you an AI-generated image with watermark.
Your task is to give some qualities based on the given watermarked image.
You need to carefully combine the image content to analyze the possible watermark features, and finally provide the type, quality score, and security/robustness scores in the required format.

For residual watermark, first indicate that the watermark type is a residual watermark, and provide a quality score (a floating-point number between 1.0 and 5.0), along with three security scores against erasure attacks (robustness indicators, each 0 or 1).
E.g.:
<type>residual watermark</type>
<quality>2.72</quality>
<jpeg>1</jpeg>

---

<gaussian>1</gaussian>
<filter>1</filter>

For a watermark-free or performance-lossless semantic watermark, first specify that the watermark type is a watermark-free or performance-lossless semantic watermark. Finally, provide a quality score (an integer from 1 to 3, where 1 = low and 3 = high) and a security score (an integer from 1 to 3, where 1 = low and 3 = high).
E.g.:
<type>watermark-free or performance-lossless semantic watermark</type>
<quality>3</quality>
<security>3</security>

For semantic watermark with ring patterns, first specify that the watermark type is semantic watermark with ring patterns. Finally, provide a quality score (an integer from 1 to 3, where 1 = low and 3 = high) and a security score (an integer from 1 to 3, where 1 = low and 3 = high).
E.g.:
<type>semantic watermark with ring patterns</type>
<quality>1</quality>
<security>1</security>

---

**User Prompt**

Please analyze the following image and determine whether it contains a residual watermark or a semantic watermark. Then, provide the type, quality score, and security/robustness scores in the required format.

## A.3. Prompts for Interpretability Cold Start, Generalization Enhancement via GRPO, and Evaluation

This subsection presents the system prompt and user prompt used in the Interpretability Cold Start and Generalization Enhancement via GRPO stages. The prompts instruct the model to first generate reasoning about the image, then determine the watermark category based on that reasoning, and finally output the corresponding quality and security scores. During the final evaluation, the same prompt is employed for both WMVLM and all baseline models to ensure a fair comparison.

---

**System Prompt**

You are a watermarked image analyzer.
I will give you an AI-generated image with watermark.
Your task is to give a thinking process and qualities based on the given watermarked image.
You need to carefully combine the image content to analyze the possible watermark features. After reasoning, provide type, quality score, and security/robustness scores in the required format. Your response should be as detailed as possible.

For residual watermark, after reasoning, indicate that the watermark type is a residual watermark, and provide a quality score (a floating-point number between 1.0 and 5.0), along with three security scores against erasure attacks (robustness indicators, each 0 or 1).
E.g.:
<think>The image presents an abstract textile-like pattern with white dotted circles and black wavy bands on a maroon background. Its visual quality is severely degraded, characterized by blurry edges, smeared details, and a pervasive, greasy-looking smudging effect across the entire canvas. These widespread artifacts indicate the presence of a strong, residual watermark that has been aggressively embedded, fundamentally altering the image's pixel data rather simply overlaying a pattern. Due to the extreme alteration and significant loss of visual fidelity, the watermark is considered highly robust. This aggressive embedding makes it very resilient and difficult to remove, ensuring it can withstand common attacks like compression, noise addition, and filtering.</think>
<type>residual watermark</type>
<quality>2.72</quality>

<jpeg>1</jpeg>
<gaussian>1</gaussian>
<filter>1</filter>

For a watermark-free or performance-lossless semantic watermark, after reasoning, specify that the watermark type is a watermark-free or performance-lossless semantic watermark. Finally, provide a quality score (an integer from 1 to 3, where 1 = low and 3 = high) and a security score (an integer from 1 to 3, where 1 = low and 3 = high).
E.g.:
<think>The image, a high-quality surreal digital painting, underwent thorough visual inspection for common watermarking artifacts such as patterns, overlays, or distortions. No visual flaws or unnatural patterns were found, and the image appeared aesthetically coherent and clean. This leads to the conclusion that the image is either watermark-free or contains a performance-lossless semantic watermark. This type of watermark is perfectly integrated into the artwork during generation, leaving no visible trace or degradation in visual quality. Consequently, the image maintains pristine visual quality and offers the highest level of security, as the watermark's complete invisibility makes it undetectable and robust against removal attempts.</think>
<type>watermark-free or performance-lossless semantic watermark</type>
<quality>3</quality>
<security>3</security>

For semantic watermark with ring patterns, after reasoning, specify that the watermark type is semantic watermark with ring patterns. Finally, provide a quality score (an integer from 1 to 3, where 1 = low and 3 = high) and a security score (an integer from 1 to 3, where 1 = low and 3 = high).
E.g.:
<think>The image is a surreal architectural scene with a painterly aesthetic. Upon close inspection of uniform areas like the stone steps and grassy textures, subtle but distinct concentric ring patterns are visible. These organized, geometric rings emanate from the image center and are inconsistent with natural textures, indicating the presence of an integrated semantic watermark. The watermark's rings introduce noticeable visual distortion and unnatural texturing, degrading the image's finer details and resulting in low visual quality. Due to the high visibility of these predictable patterns, the watermark offers low security, as it is easily detectable and potentially vulnerable to removal by a trained observer.</think>
<type>semantic watermark with ring patterns</type>
<quality>1</quality>
<security>1</security>

User Prompt

Please analyze the following image and determine whether it contains a residual watermark or a semantic watermark. Then, after reasoning, provide the type, quality score, and security/robustness scores in the required format.

## B. Evaluation Examples

In this section, we provide concrete examples of WMVLM's evaluation outputs, as illustrated in Fig 3 and Fig 4. These examples demonstrate the model's ability to produce structured reasoning and accurate quantitative assessments for different watermarking paradigms.

For each example, we show:

- Input image: The watermarked image provided to the model.

- Reasoning text: The model's step-by-step analysis enclosed in <think></think> tags.

- Structured output: The final prediction formatted with <type></type>, <quality></quality>, and <security></security> (or <jpeg></jpeg>, <gaussian></gaussian>, and <filter></filter> for residual watermarks) tags.

The examples include both residual watermarks (where the model outputs a continuous quality score and three binary robustness indicators) and semantic watermarks (where quality and security are given as discrete ordinal scores). These results demonstrate WMVLM's ability to combine interpretable textual reasoning with precise quantitative scoring, thereby providing a unified and comprehensive evaluation framework.

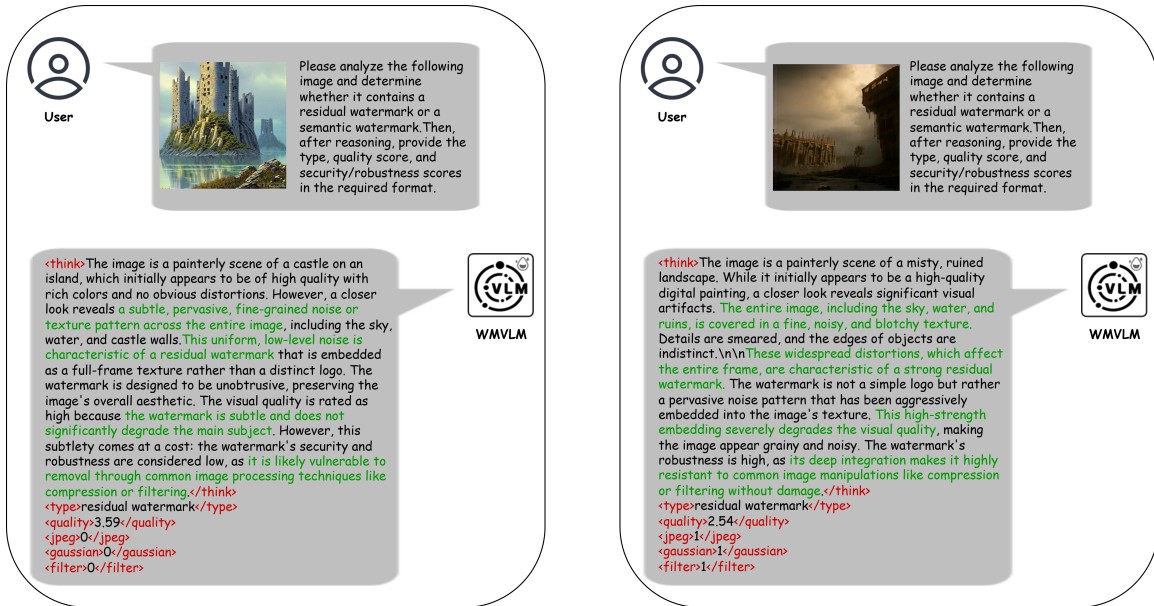

*Figure 3.* Examples of residual watermarks evaluation.

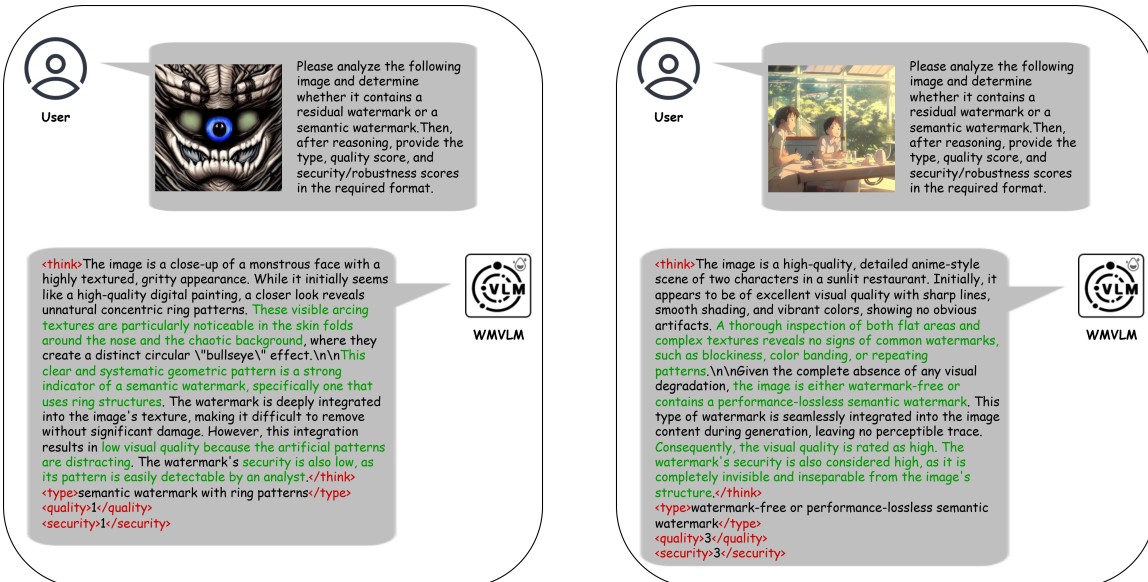

*Figure 4.* Examples of semantic watermarks evaluation.

## C. Visualization of Model Perception

To provide a deeper understanding of WMVLM's internal perception and analysis mechanism, we visualize the model's attention maps when processing different watermarking paradigms. Unlike conventional classification models that make discrete decisions, WMVLM is designed to perceive watermark artifacts from a watermarked image, analyze their spatial and textural characteristics, and generate an interpretable report that identifies suspicious watermark features along with

corresponding quality and security assessments. Visualizing attention maps allows us to examine which image regions the model focuses on when producing its analysis, thereby revealing whether the model captures the correct watermark features. Three representative watermarking methods are selected for visualization: two residual watermarking methods, HiDDeN (Zhu et al., 2018) and VINE (Lu et al., 2025), and one semantic watermarking method, RingID (Ci et al., 2024), which embeds ring patterns in the latent space.

### C.1. Attention Map Extraction

The attention maps are extracted from the final layer of WMVLM, specifically focusing on the attention weights corresponding to visual tokens. For each input image, we compute the mean attention across all attention heads for the last token, which captures the model's focus when producing the final prediction.

For visualization, we adopt different layout strategies for different watermark types. Specifically, for residual watermarking methods HiDDeN and VINE, each figure is organized from left to right as three columns: the original watermarked image, WMVLM's attention map, and the watermark residual map, which is defined as the difference between the watermarked and original images. For the semantic watermarking method RingID, which lacks a visually meaningful residual map, we present two columns: the original watermarked image and WMVLM's attention map.

### C.2. Residual Watermarks: HiDDeN and VINE

As shown in Fig. 5, HiDDeN embeds watermarks by spreading residuals diffusely across the entire image. Consequently, WMVLM's attention is also broadly distributed, without focusing on any specific local region. In contrast, Fig. 6 demonstrates that VINE's watermark characteristics are predominantly concentrated near image edges. Remarkably, WMVLM's attention map closely follows this distribution, focusing sharply on edge regions where watermark information is most salient.

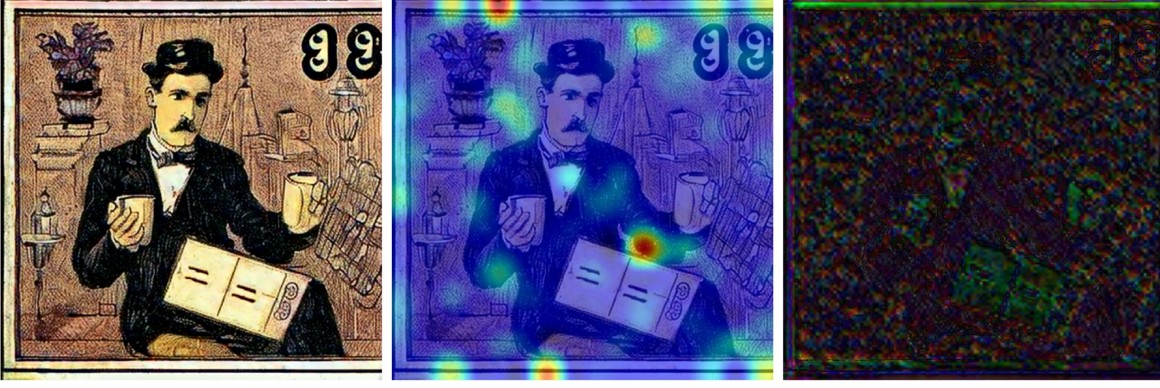

*Figure 5.* Visualization of HiDDeN. From left to right: watermarked image, attention map, and watermark residual map.

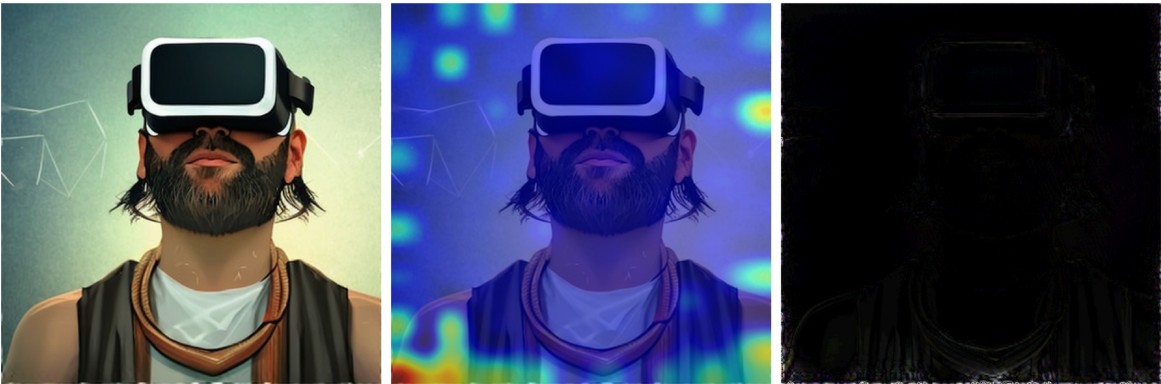

*Figure 6.* Visualization of VINE. From left to right: watermarked image, attention map, and watermark residual map.

## C.3. Semantic Watermark: RingID

For semantic watermarks, RingID embeds watermark messages by introducing ring-shaped structures in the latent space. If such structures were directly visible in the pixel space, detecting the watermark would be relatively straightforward. The more challenging case is whether the model can still identify the watermark when no explicit ring pattern can be visually observed in the image. As shown in Fig. 7, WMVLM concentrates its attention on the central region, suggesting that it uses this region to extract discriminative watermark-related features. Rather than indicating a visible ring in the pixel space, the attention map shows where WMVLM obtains strong feature responses associated with RingID. This suggests that WMVLM can act as an effective feature extractor, capturing features that are consistent with latent or frequency-domain ring signatures even when the corresponding pixel-level evidence is weak or visually implicit.

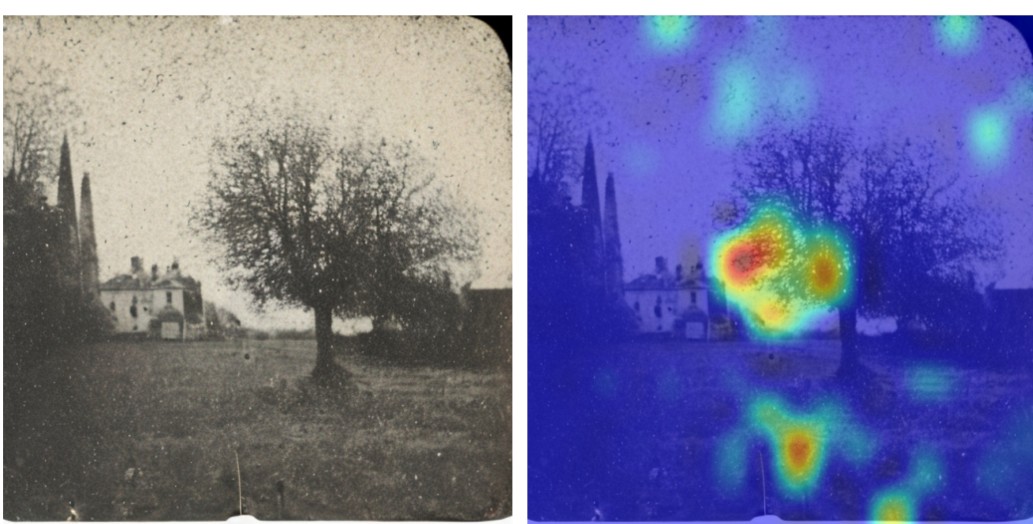

*Figure 7.* Visualization of RingID. From left to right: watermarked image and attention map.

## C.4. Summary

Collectively, these attention map visualizations demonstrate a strong alignment between WMVLM's attentional focus and the underlying watermark characteristics. For residual watermarks with spatially distributed or edge-concentrated residuals, the model's attention faithfully mirrors these distributions. For semantic watermarks with latent-space ring patterns, the model successfully attends to the corresponding image regions despite the absence of visual artifacts. These observations confirm that WMVLM captures the correct watermark features and that the visual evidence provided by attention maps is in strong agreement with the model's analytical outputs, validating WMVLM's ability to perform unified, interpretable watermark assessment.

# D. Data Preparation

In this section, we first describe the dataset partitioning scheme and then elaborate on the specific definitions of quality and security scores for both residual and semantic watermarks.

## D.1. Dataset Partitioning

To construct the dataset, we include 9 representative watermarking methods: 6 residual watermarks including Dwt-Dct (Cox et al., 2007), RivaGAN (Zhang et al., 2019), HiDDeN (Zhu et al., 2018), RobustWide (RW) (Hu et al., 2024b), VINE (Lu et al., 2025), and Stable Signature (SS) (Fernandez et al., 2023), along with three semantic watermarks including RingID (Ci et al., 2024), Tree-Ring (Wen et al., 2023), and a unified category of performance-lossless (Lossless) techniques comprising Gaussian Shading (GS) (Yang et al., 2024), PRCW (Gunn et al., 2025), and Gaussian Shading++ (GS++) (Yang et al., 2025b). All images are generated using SD v2.1 (Rombach et al., 2022) with prompts sourced from the Stable-Diffusion-Prompts-2.47M (FredZhang7, 2024), yielding 5,000 images for residual watermarks (denoted as $\mathcal{D}_i^{(\text{res})}$) and 9,500 images for semantic watermarks (denoted as $\mathcal{D}_j^{(\text{sem})}$). For each watermarking method, 500 images are

reserved for testing, with the remaining images used for training.

We adopt different dataset splits for the three training stages. In the category and score pre-training stage, we use the first 4,500 images of each residual watermark as the training set ($\mathcal{D}_i^{(\text{res})}[: 4500]$) and the remaining 500 images as the test set ($\mathcal{D}_i^{(\text{res})}[4500 :]$). Similarly, for semantic watermarks, the first 9,000 images constitute the training set ($\mathcal{D}_j^{(\text{sem})}[: 9000]$) and the last 500 images form the test set ($\mathcal{D}_j^{(\text{sem})}[9000 :]$). This split ensures a comparable total number of training images between residual and semantic watermarks, preventing bias during pre-training.

During the cold start stage, we generate 500 interpretable text samples per method using Gemini-2.5-Pro (Comanici et al., 2025). The corresponding images for these explanations are $\mathcal{D}_i^{(\text{res})}[: 500]$ for residual watermarks and $\mathcal{D}_j^{(\text{sem})}[: 500]$ for semantic watermarks, while the test sets remain $\mathcal{D}_i^{(\text{res})}[4500 :]$ and $\mathcal{D}_j^{(\text{sem})}[9000 :]$, respectively.

In the generalization enhancement via GRPO stage, we use 1,000 non-overlapping images per method as the training set ($\mathcal{D}_i^{(\text{res})}[500 : 1500]$ and $\mathcal{D}_j^{(\text{sem})}[500 : 1500]$) to promote exploration during reinforcement learning. The test sets remain unchanged from the previous stages, i.e., $\mathcal{D}_i^{(\text{res})}[4500 :]$ and $\mathcal{D}_j^{(\text{sem})}[9000 :]$.

### D.2. Residual Scores

The assessment of visual quality for residual watermarks has been well-established, with PSNR being the most classic metric, as it effectively reflects the intensity of watermark-induced residual distortion. Consequently, we adopt PSNR as the basis for evaluating the visual quality of residual watermarks. The PSNR values of all watermarked images are calculated and normalized to a floating-point score within the range of [1, 5], denoted as $v^{(\text{res})} \in [1, 5]$.

Due to the inherent vulnerability of residual watermarks against deep learning-based erasure attacks (Zhao et al., 2024), we focus on their robustness to classical OSNs interference for security evaluation. Specifically, three typical types of noise are considered: JPEG compression, Gaussian noise, and median filtering. Based on whether the watermark can withstand each corresponding noise attack, the security assessment is encoded into three independent binary classification labels, denoted as $s^{(\text{res})} = (s_j, s_g, s_f) \in \{0, 1\}^3$, with $s_j$, $s_g$, and $s_f$ representing robustness against JPEG compression, Gaussian noise, and median filtering, respectively. For each method, we select 1,000 images and conduct experiments on six residual watermarking techniques: DwtDct (Cox et al., 2007), RivaGAN (Zhang et al., 2019), HiDDeN (Zhu et al., 2018), RobustWide (RW) (Hu et al., 2024b), VINE (Lu et al., 2025), and Stable Signature (SS) (Fernandez et al., 2023). The robustness results are presented in Tab.6. We set an 85% threshold to determine whether a watermark can resist a given type of noise, and the corresponding security scores are also reported in Tab.6.

*Table 6.* Robustness of residual watermarks under three common OSNs noises and corresponding security scores.

| Methods | Accuracy | | | Security Scores | | |
|---|---|---|---|---|---|---|
| | JPEG | Gaussian Noise | Median Filter | JPEG | Gaussian Noise | Median Filter |
| DwtDct | 0.512 | 0.743 | 0.537 | 0 | 0 | 0 |
| RivaGAN | 0.965 | 0.925 | 0.980 | 1 | 1 | 1 |
| HiDDeN | 0.882 | 0.864 | 0.885 | 1 | 1 | 1 |
| RW | 1.000 | 0.934 | 1.000 | 1 | 1 | 1 |
| VINE | 1.000 | 0.959 | 1.000 | 1 | 1 | 1 |
| SS | 0.901 | 0.946 | 0.939 | 1 | 1 | 1 |

### D.3. Semantic Watermark Scores

Semantic watermarks operate by altering the distribution of latent representations to convey watermark messages. Recent studies (An et al., 2024; Lin & Juarez, 2025) have revealed that the distribution shift introduced by watermark embedding not only leads to a degradation in visual quality but also functions as a recognizable signature, making it susceptible to adversarial erasure attacks. Therefore, our analysis bypasses the image level and directly examines the latent representation space. For each semantic watermarks method, we generate a substantial set of watermarked latent representation samples. Multiple hypothesis tests, which include the Cramér-von Mises test (Cramér, 1928; Von Mises, 1936), Jarque-Bera test (Jarque & Bera, 1987), and D'Agostino's K² test (D'agostino et al., 1990), are employed to compute the $p$-value between the watermarked latent distribution and the standard normal distribution. This $p$-value serves as a measure of distributional

deviation. A higher $p$-value indicates that the watermarked latent distribution is closer to the standard normal distribution, corresponding to higher perceived image quality and security. Based on the statistical characteristics observed in prevailing semantic watermarking methods, both the quality and security scores are categorized into a three-level discrete scale, denoted as $v^{\text{(sem)}} \in \{1, 2, 3\}$ and $s^{\text{(sem)}} \in \{1, 2, 3\}$.

Specifically, for each selected semantic watermarking method, we generate 80,000 latent representation samples and concatenate them into a one-dimensional tensor to measure their deviation from the standard normal distribution. The resulting $p$-values for all methods are presented in Tab.7. Based on their performance across the hypothesis tests, we assign both quality and security scores according to a three-level discrete scale: RingID (Ci et al., 2024), which yields poor results across all three tests, is labeled as 1 (weak); Tree-Ring (Wen et al., 2023) and GaussianMarker (Li et al., 2025a), which show relatively low $p$-values under the Cramér–von Mises test, are labeled as 2 (moderate); while GS (Yang et al., 2024), PRCW (Gunn et al., 2025), GS++ (Yang et al., 2025b), T2SMark (Yang et al., 2025a), and the unwatermarked images (Origin Image), which perform well across all three tests, are labeled as 3 (strong).

*Table 7.* Hypothesis test results of semantic watermarks and corresponding quality and security scores.

| Methods | Hypothesis Test ($p$-value $\uparrow$) | | | Scores | |
|---|---|---|---|---|---|
| | Cramér-von Mises | Jarque-Bera | D'Agostino's K² | Quality | Security |
| RingID | 4.877e-06 | 9.841e-46 | 9.552e-46 | 1 | 1 |
| Tree-Ring | 2.575e-07 | 7.265e-02 | 7.236e-02 | 2 | 2 |
| GaussMarker | 2.892e-07 | 7.023e-02 | 6.492e-02 | 2 | 2 |
| GS | 3.054e-01 | 6.820e-02 | 6.914e-02 | 3 | 3 |
| PRCW | 1.744e-01 | 2.050e-02 | 2.052e-02 | 3 | 3 |
| GS++ | 4.166e-01 | 9.786e-01 | 9.786e-01 | 3 | 3 |
| T2SMark | 2.768e-01 | 6.812e-02 | 6.841e-02 | 3 | 3 |
| Origin Image | 2.317e-01 | 3.860e-02 | 3.860e-02 | 3 | 3 |

# E. Additional Ablation Studies

In this section, we present ablation results for key hyperparameters, as shown in Tab. 8. The **Default** parameter configuration is as follows: $r^k_{\text{fmt\_penalty}} = -10$, $G = 8$, $\beta = 0.01$. All other experimental settings remain consistent with the **Default** unless otherwise specified in the table.

### E.1. Format Error Penalty $r^k_{\text{fmt\_penalty}}$

In the original experimental setup, we assigned a reward of 0 for correct template compliance and a penalty of -10 for format errors. In the ablation experiments, we tested milder penalties of -1 and -5. The results show that overly lenient format penalties allow the model to deviate from the required output structure, leading to evaluation failures, including misclassification of watermark types and inaccurate quality/security scores. This effect is particularly evident in the PLCC results for RivaGAN (Zhang et al., 2019) and HiDDeN (Zhu et al., 2018). These findings confirm the effectiveness of applying a stronger penalty in WMVLM's unified and complex evaluation framework.

### E.2. Group Size $G$

The group size $G$ controls the number of responses sampled per prompt in GRPO. Results indicate that $G = 8$ yields the best overall performance, suggesting that increasing the group size benefits model exploration. For instance, the PLCC results for RivaGAN (Zhang et al., 2019) and HiDDeN (Zhu et al., 2018) show that broader exploration helps the model better capture the intrinsic relationship between quality scores and watermark features, leading to more effective evaluation. This demonstrates the advantage of setting a larger group size in our framework.

### E.3. KL-Divergence Penalty $\beta$

The divergence penalty $\beta$ controls the magnitude of the KL-Divergence penalty in GRPO training, i.e., the extent to which the policy $\pi_\theta$ may diverge from the SFT reference model $\pi_{\text{ref}}$. Ablation results indicate that $\beta = 0.1$ yields the best overall

performance, demonstrating that imposing a stronger KL-Divergence penalty during GRPO training is beneficial. When $\beta$ is set too small, the model is permitted to explore more freely, which can lead to the forgetting of previously acquired templates and knowledge. This effect is particularly pronounced at $\beta = 0$, where unrestricted exploration results in training collapse and a complete failure to perform the evaluation task. In summary, for WMVLM that is designed to execute unified and complex evaluations, applying a sufficiently large KL-Divergence penalty during GRPO training is essential.

*Table 8.* Ablation study on key hyperparameters of WMVLM. Performance is reported as PLCC / SRCC / Average security accuracy (three noise types) for residual watermarks, and as Quality accuracy / Security accuracy for semantic watermarks. **Bold** represents the best, underline represents the second best.

| Hyperparameters | Residual Watermarks | | | | | | Semantic Watermarks | | |
|---|---|---|---|---|---|---|---|---|---|
| | DwtDct | RivaGAN | HiDDeN | RW | VINE | SS | RingID | Tree-Ring | Lossless |
| $r^k_{\text{fmt\_penalty}} = -1$ | **0.952/0.958**/0.874 | 0.856/0.869/0.858 | 0.388/0.606/0.878 | 0.965/0.970/0.933 | **0.983**/0.984/0.962 | 0.980/0.983/0.960 | 0.726/0.726 | 0.934/0.934 | 0.882/0.882 |
| $r^k_{\text{fmt\_penalty}} = -5$ | 0.892/0.957/0.988 | 0.178/0.850/0.984 | 0.438/0.599/0.990 | 0.964/**0.974**/0.994 | 0.940/0.971/0.994 | 0.956/0.975/0.996 | 0.824/0.824 | 0.964/0.964 | 0.902/0.902 |
| $G = 4$ | 0.950/0.946/0.982 | 0.215/0.870/0.984 | 0.418/0.627/**1.000** | **0.973/0.974**/0.990 | 0.950/0.975/0.996 | 0.965/0.984/**0.998** | 0.826/0.826 | 0.966/0.966 | **0.918/0.918** |
| $G = 6$ | **0.952/0.958**/0.990 | 0.177/0.871/0.974 | 0.330/0.605/**1.000** | 0.950/0.972/**0.994** | 0.966/0.977/0.996 | **0.982/0.985**/0.994 | 0.822/0.822 | **0.968/0.968** | 0.906/0.906 |
| $\beta = 0.0$ | 0.916/0.886/0.046 | None/None/None | None/None/None | None/None/None | None/None/None | None/None/None | None/None | None/None | None/None |
| $\beta = 0.05$ | 0.949/0.953/0.972 | **0.892/0.876**/0.948 | 0.487/0.616/0.956 | 0.961/0.968/0.974 | 0.979/0.983/0.970 | 0.979/0.981/0.939 | 0.717/0.717 | 0.944/0.944 | 0.858/0.858 |
| **Default** | **0.952**/0.957/0.988 | 0.885/0.875/**0.990** | **0.712/0.708**/0.998 | 0.961/0.973/**0.994** | **0.983/0.986/1.000** | 0.981/**0.985/0.998** | **0.852/0.852** | 0.956/0.956 | 0.910/0.910 |

