# OpenReview forum: "WMVLM: Evaluating Diffusion Model Image Watermarking via Vision-Language Models"
_ICML.cc/2026/Conference — ICML 2026 regular_

### Official Review · Reviewer_mTUV · 2026-02-24

**Soundness:** 3
**Presentation:** 3
**Significance:** 3
**Originality:** 3
**Overall Recommendation:** 5
**Confidence:** 2

**Summary:**

The authors propose WMVLM, the first unified and interpretable evaluation framework for diffusion model image watermarking. To establish this framework, the authors redefine the quality and security evaluation metrics for watermarks. Furthermore, they employ a three-stage training strategy to fine-tune and align the VLM; specifically, this entails classification and scoring pre-training, an interpretable cold start, and enhanced generalization via GRPO.

**Compliance With Llm Reviewing Policy:**

Affirmed.

**Final Justification:**

The author addressed my concerns and questions; I am willing to give them a higher rating.

**Key Questions For Authors:**

No.

**Limitations:**

Yes.

**Strengths And Weaknesses:**

Strengths

1. The authors propose WMVLM, which, to the best of my knowledge, is the first unified and interpretable evaluation framework utilizing VLMs to assess both residual and semantic watermarks. By doing so, the authors successfully bridge a significant gap in the literature, as these two paradigms previously required entirely disparate evaluation metrics.

2. The authors design a logical and highly effective three-stage training pipeline. Their progression from SFT for category/score alignment, to a Knowledge Distillation-based "cold start" for interpretability, and finally to GRPO for generalization enhancement, is technically sound and well-justified by the ablation studies.

3. The authors conduct comprehensive comparisons against 11 state-of-the-art VLMs (including closed-source models like GPT-5 and Gemini-3-Pro, and open-source models like Qwen3-VL and Gemma-3). Through these experiments, the authors demonstrate that WMVLM achieves overwhelming superiority in both quantitative accuracy and the generation of interpretable, human-readable reasoning.

Weaknesses

Disclaimer: I must candidly disclose that my primary research expertise lies outside the specific subfield of digital watermarking for diffusion models. Consequently, I may not be equipped to provide deep, domain-specific critiques regarding the fundamental mathematical formulations of the watermarking algorithms themselves. However, from a broader machine learning and model evaluation perspective, I raise the following primary concern:

1. While the authors present cross-model generalization results (testing on SD v1.4 after training on SD v2.1), their evaluation remains heavily constrained to the SD family. To convincingly demonstrate that WMVLM is a truly universal and robust evaluation framework, the authors must expand their testing beyond SD. I strongly encourage the authors to include experiments evaluating watermarked images generated by a more diverse set of state-of-the-art models, specifically: Nano-Banana, Z-Image, Qwen-Image, FLUX 2.

---

> ### Author Rebuttal · Authors · 2026-03-31
>
> We sincerely thank the reviewer for the candid and constructive feedback. We appreciate the recognition of our work from a broader machine learning perspective. To address the concern regarding generalization across diverse model families, we have conducted extensive additional experiments.
>
> We evaluated four representative models to assess WMVLM’s cross-architecture performance: Nano-Banana (closed-source), Z-Image, Qwen-Image, and FLUX.2 (open-source). Given the restricted timeframe, we focused on evaluating the zero-shot generalization of our current WMVLM (trained on SD v2.1) and omitted methods requiring per-model fine-tuning (e.g., Stable Signature). The zero-shot generalization results of WMVLM (trained on SD v2.1) are as follows:
>
> |  | DwtDct | RivaGAN | HiDDeN | RW | VINE | RingID | Tree-Ring | Lossless |
> | --- | --- | --- | --- | --- | --- | --- | --- | --- |
> | Nano-Banana | 0.914/0.903/0.984 | 0.771/0.766/0.992 | 0.610/0.612/0.998 | 0.935/0.922/0.82 | 0.982/0.983/0.996 | - | - | - |
> | Z-Image | 0.957/0.957/0.982 | 0.766/0.752/0.992 | 0.606/0.613/0.996 | 0.948/0.925/0.986 | 0.980/0.980/0.992 | 0.332/0.332 | 0.354/0.354 | 0.896/0.896 |
> | Qwen-Image | 0.930/0.994/0.994 | 0.841/0.809/0.994 | 0.653/0.641/1.000 | 0.956/0.941/0.988 | 0.989/0.992/1.000 | 0.340/0.340 | 0.328/0.328 | 0.892/0.892 |
> | FLUX2 | 0.907/0.909/0.928 | 0.739/0.668/0.988 | 0.578/0.577/0.996 | 0.881/0.928/0.970 | 0.982/0.985/0.996 | 0.304/0.304 | 0.302/0.302 | 0.904/0.904 |
> | SD V2.1 | 0.952/0.957/0.988 | 0.850/0.875/0.990 | 0.712/0.708/0.998 | 0.961/0.973/0.994 | 0.983/0.986/1.000 | 0.852/0.852 | 0.956/0.956 | 0.910/0.910 |
>
> Our results show that WMVLM generalizes remarkably well to residual watermarks across all families. In contrast, for semantic watermarks (RingID and Tree-Ring), we observe a performance gap when transitioning to newer architectures. We attribute this to fundamental shifts in latent space:
> 1.  **Signal Dilution:** While our base model (SD v2.1) operates on a [B, 4, H, W] latent space, advanced models utilize a [B, 16, H, W] structure. Even when maintaining a constant embedding ratio (e.g., 4/16 vs. 1/4 channels), the significantly larger latent volume tends to **dilute the watermark's perceptual signature**.
>
> 1. **Structural Transformation:** Crucially, many modern architectures reshape latent channels during inference (e.g., into flattened [B, D, L] sequences), which fundamentally alters the **spatial-structural manifestation** of the watermark that the VLM was initially trained to recognize.
>
> To bridge this gap, we conducted a lightweight adaptation using only 1,000 samples of Tree-Ring and RingID watermarks generated by Z-Image for 5 epochs of GRPO fine-tuning. Results are as follows:
>
> |  | RingID | Tree-Ring |
> | --- | --- | --- |
> | Finetuned Model | 0.676/0.676 | 0.704/0.704 |
> | WMVLM | 0.332/0.332 | 0.354/0.354 |
>
> Experimental results demonstrate that such rapid fine-tuning yields substantial performance gains, effectively elevating WMVLM’s recognition capabilities on previously unseen architectures. While these preliminary accuracy levels do not yet match the performance on the original SD v2.1 dataset, this lightweight adaptation successfully confirms the VLM's inherent capacity to capture the underlying watermark features in the latest generative models. These results serve as a proof-of-concept: with sufficient time and computational resources to reconstruct a large-scale dataset across diverse model families and execute the full training pipeline, we anticipate that further improvements in robustness and fidelity are achievable.

---

> > ### Author Rebuttal · Reviewer_mTUV · 2026-04-01
> >
> > The author addressed my concerns and questions; I am willing to give them a higher rating.

---

> > > ### Author Response · Authors · 2026-04-05
> > >
> > > Thank you for your careful review, constructive feedback, and for updating your score. We greatly value your time, effort, and support in strengthening our paper.

---

### Official Review · Reviewer_L8Ey · 2026-03-10

**Soundness:** 3
**Presentation:** 4
**Significance:** 4
**Originality:** 3
**Overall Recommendation:** 6
**Confidence:** 5

**Summary:**

This paper presents WMVLM, the first unified and interpretable evaluation framework for diffusion model image watermarking based on VLMs. To address the limitations of existing methods—such as the lack of a unified framework, absence of interpretability, insufficient security considerations, and inappropriate metrics for semantic watermarks—the authors redefine quality and security metrics for both residual watermarks and semantic watermarks. Furthermore, the paper introduces a three-stage training strategy comprising classification and scoring pre-training, interpretability cold-start, and GRPO-based generalization enhancement to enable the model to simultaneously output quantitative scores and qualitative interpretable text.

**Compliance With Llm Reviewing Policy:**

Affirmed.

**Final Justification:**

The proposed WMLLM method effectively redefines the quality and security metrics for both residual and semantic watermarks. Given its strengths and the authors' satisfactory rebuttal, I recommend accepting this work.

**Key Questions For Authors:**

1.Have the authors compared the impact of different model scales (e.g., 4B vs. 8B) on evaluation accuracy? Considering inference latency for large-scale datasets, what is the specific trade-off between WMVLM's superior accuracy and its higher computational cost compared to traditional metrics ?

2.Since the SFT stage requires a substantial amount of data, how many fine-tuning samples would be needed for WMVLM to maintain its current robustness when faced with entirely new watermarking principles in the future?

**Limitations:**

Yes

**Strengths And Weaknesses:**

Strengths

1.This paper pioneers the task of evaluating watermarked image quality using VLMs. This approach is highly innovative and establishes a promising new direction for the field of watermark quality assessment.
2.WMVLM is a unified and reference-free evaluation framework that enables the assessment of both quality and security for semantic and residual watermarks.
3.The proposed evaluation metrics for semantic watermarks are novel. By transcending pixel-level analysis to delve into the latent space, the authors effectively quantify image quality through the lens of distribution shifts, providing a more accurate assessment.
4.The training strategy is efficient, progressing from SFT-based watermarking feature learning and interpretability cold-start to the integration of GRPO for generalization enhancement. This pipeline ensures the model's robust performance in both quantitative scoring and qualitative reasoning.
5.The paper is logically structured and well-written. The Fig.2 effectively assists the reader in understanding the complex training process.

Weaknesses

1.Although the three-stage training strategy is conceptually efficient, the initial SFT stage remains highly dependent on the construction of large-scale, high-quality annotated datasets. The collection and labeling of scores for diverse watermarking paradigms can be time-consuming and labor-intensive, which may limit the framework's agility when adapting to entirely new watermarking technologies.

2.The framework utilizes an 8B-parameter model; however, inference latency is a critical factor in large-scale evaluation scenarios. It remains unclear whether such a large model scale is strictly necessary. If a smaller model (e.g., Qwen3-4B) could achieve comparable performance, the current 8B choice might represent an inefficient use of computational resources.

3.There are some minor presentation issues in the figures. For instance, in Fig.2, the superscript "(res)" is incorrectly typeset in italics in several places.

---

> ### Author Rebuttal · Authors · 2026-03-31
>
> We sincerely thank the reviewer for the highly positive evaluation of our work's presentation and significance, and for the expert insights regarding the scalability and efficiency of WMVLM. We have conducted additional experiments to address your specific concerns.
>
> **1. W1: Efficiency of SFT and Data Agility**
>
> We agree that data construction is a key stage. However, we would like to clarify that **the labeling process in WMVLM is entirely automated**, utilizing our evaluation pipeline to generate scores without human intervention. To further address your concern about adapting to "entirely new technologies," we conducted a stress test (see **Q2** below). The results show that WMVLM can adapt to new paradigms with **minimal data (1,000 samples) and low training cost (4 hours)**, demonstrating high practical agility.
>
> **2. W2: Necessity of 8B-Parameter Model**
>
> To address the concern regarding whether an 8B model is strictly necessary, we conducted a comparative experiment using **Qwen3-VL-4B-Instruct** as the backbone, following the identical training pipeline.
>
> |  | DwtDct | RivaGAN | HiDDeN | RW | VINE | SS | RingID | Tree-Ring | Lossless |
> | --- | --- | --- | --- | --- | --- | --- | --- | --- | --- |
> | Qwen3-VL-4B-Instruct | 0.865/0.944/0.982 | 0.841/0.839/0.988 | 0.670/0.661/0.990 | 0.829/0.951/0.988 | 0.958/0.977/0.994 | 0.964/0.965/**1.000** | 0.844/0.844 | 0.900/0.900 | 0.854/0.854 |
> | Qwen3-VL-8B-Instruct (WMVLM) | **0.952**/**0.957**/**0.988** | **0.850**/**0.875**/**0.990** | **0.712**/**0.708**/**0.998** | **0.961**/**0.973**/**0.994** | **0.983**/**0.986**/**1.000** | **0.981**/**0.985**/0.998 | **0.852**/**0.852** | **0.956**/**0.956** | **0.910**/**0.910** |
>
> The results indicate that while the 4B model is functional, the 8B version provides a **substantial performance leap**. In the context of a unified evaluation framework where precision is paramount, we believe the 8B model serves as a necessary foundation to ensure reliable and high-fidelity scoring across diverse watermarking paradigms.
>
> **3. W3: Presentation Improvements**
>
> We appreciate your meticulous review. We have corrected the italicization of the superscript **"(res)"** in Figure 2 and are conducting a full-text audit to ensure all notation adheres to standard typesetting conventions.
>
>
> **4. Q1: Trade-offs between Superior Accuracy and Computational Costs**
>
> Regarding the trade-off between WMVLM’s costs and traditional metrics, we emphasize three unique advantages:
>
> - **No-Reference (NR) Capability:** Traditional high-quality metrics (PSNR, SSIM, LPIPS) are **reference-based**, requiring the original ground-truth image. In real-world forensic or detection scenarios, original images are typically unavailable. WMVLM’s NR nature provides a practical utility that pixel-level metrics cannot match.
> - **Unique Interpretability:** Beyond providing a numerical score, WMVLM offers **qualitative reasoning** (e.g.,  how a watermark affects image texture). This diagnostic value is a significant advancement over black-box traditional metrics, justifying the additional inference latency in professional audit scenarios.
> - **Efficiency vs. Global Metrics:** While slower than pixel metrics, WMVLM is more efficient for certain tasks than metrics like FID, which require large reference batches and complex feature modeling.
>
> **5. Q2: Robustness against New Principles with Minimal Data**
>
> While our initial training stages benefit from a comprehensive dataset, WMVLM shows encouraging data efficiency when adapting to previously unseen watermarking principles. To explore this, we conducted preliminary experiments on two residual watermarks, MBRS [1] and FIN [2]. Notably, with 1,000 samples and 4 epochs of GRPO fine-tuning, the model demonstrated a measurable improvement over its initial zero-shot performance. These results suggest that WMVLM can serve as a practical starting point for evaluating emerging technologies with relatively low data requirements.
>
> |  | MBRS | FIN |
> | --- | --- | --- |
> | Finetuned Model | 0.884/0.899/0.878 | 0.761/0.770/0.832 |
> | WMVLM | 0.520/0.844/0.398 | 0.493/0.716/0.506 |
>
> [1] Jia, Zhaoyang, et al. "MBRS: Enhancing robustness of dnn-based watermarking by mini-batch of real and simulated jpeg compression." ACM MM. 2021.
>
> [2] Fang, Han, et al. "Flow-based robust watermarking with invertible noise layer for black-box distortions."  AAAI. 2023.

---

> > ### Author Rebuttal · Reviewer_L8Ey · 2026-04-01
> >
> > I thank the authors for their comprehensive rebuttal. The proposed WMLLM method effectively redefines quality and security metrics for both residual and semantic watermarks. The capabilities of generative models are advancing rapidly. However, evaluation methods for watermarks in generated content remain scarce. This research addresses a critical gap in the field. Consequently, the proposed method provides an undeniable contribution to society. In light of these considerations, I have decided to raise my score.

---

> > > ### Author Response · Authors · 2026-04-05
> > >
> > > Thank you so much for your thoughtful review, your valuable feedback, and for updating your score. We truly appreciate your time, effort, and support of our work.

---

### Official Review · Reviewer_26Ro · 2026-03-12

**Soundness:** 3
**Presentation:** 3
**Significance:** 3
**Originality:** 3
**Overall Recommendation:** 4
**Confidence:** 2

**Summary:**

The paper proposes WMVLM, a unified and interpretable framework for diffusion model image watermarking based on vision-language models (VLMs). The model is trained through three different stages, to enable efficient distinction between watermark features and generalization across watermarks. To support this, the paper introduces quality and security scores that quantify these properties in existing watermarks and use them as signals during training.

**Compliance With Llm Reviewing Policy:**

Affirmed.

**Final Justification:**

The rebuttal was able to address my main concerns and I recommend acceptance.

**Key Questions For Authors:**

1. How well does the proposed framework generalize to other watermarks that were not in the training set, such as TrustMark, StegaStamp and SFWMark?

2. How well does WMVLM perform against a baseline that was fine-tuned on the constructed dataset?

3. Why does the robustness evaluation focus on JPEG compression, Gaussian noise and median filtering and not other, or more, augmentation methods?

4. Why was PSNR chosen as a quality metric for the residual watermarks and not LPIPS or SSIM or a combination of the three?

**Limitations:**

yes

**Strengths And Weaknesses:**

### Strengths:

- The paper is well structured and easy to follow.
- The reward design is well aligned with the task objective and efficiently integrates multiple evaluation aspects.
- The paper provides insightful ablation studies on the training strategy of WMVLM.

### Weaknesses:

- The paper proposes a unified and interpretable evaluation framework, yet the performance is primarily evaluated on the watermarks in the training set. To verify the generalization capabilities of the framework, the evaluation should be expanded to additional watermarks, such as TrustMark [1], StegaStamp [2] and SFWMark [3].
- The paper states that “VLMs have limited zero-shot capacity for detecting watermark features and require structured supervision to effectively learn categorization and scoring tasks.” (line 203-205). However, the evaluation of WMVLM is performed against zero-shot baselines. It would strengthen the evaluation to include at least one of the baselines that is supervisedly fine-tuned on the dataset.

**References**

[1] Tu Bui, Shruti Agarwal, and John Collomosse. “TrustMark: Robust Watermarking and Watermark Removal for Arbitrary Resolution Images”. ICCV. 2025

[2] Matthew Tancik, Ben Mildenhall, and Ren Ng. “StegaStamp: Invisible Hyperlinks in Physical Photographs.”CVPR. 2020

[3] Sung Ju Lee and Nam Ik Cho. “Semantic Watermarking Reinvented: Enhancing Robustness and Generation Quality with Fourier Integrity”. ICCV. 2025

---

> ### Author Rebuttal · Authors · 2026-03-31
>
> We sincerely thank the reviewer for the thoughtful comments and the suggested additional benchmarks. We have addressed the concerns regarding generalization, baselines, and evaluation metrics through new experiments and detailed technical justifications.
>
> **1. W1 & Q1: Generalization to Additional Watermarks (TrustMark, StegaStamp, and SFW)**
>
> We appreciate the suggestion to include more diverse watermarking paradigms. Due to the limited rebuttal window, we encountered practical constraints: TrustMark[1] implementation encountered reproducibility issues, and StegaStamp[2] lacks pre-trained weights. To provide a fair evaluation, we selected MBRS [4] and FIN [5] (as representative residual watermarks) and SFW [3] (as a semantic watermark) for zero-shot testing:
>
> |  | MBRS | FIN | SFW |
> | --- | --- | --- | --- |
> | WMVLM | 0.520/0.844/0.398 | 0.493/0.716/0.506 | 0.896/0.896 |
>
> WMVLM generalizes excellently to the SFW. The lower zero-shot performance on MBRS and FIN is attributed to the distinct residual patterns compared to our training set.
>
> However, we demonstrate that WMVLM can bridge this gap with minimal data. Using only 1,000 samples for each method and 4 hours (4 epochs) of GRPO fine-tuning, the performance improved significantly. This suggests that WMVLM can serve as a flexible foundation for evaluating unseen watermarking methods.
>
> |  | MBRS | FIN |
> | --- | --- | --- |
> | Finetuned Model | 0.884/0.899/0.878 | 0.761/0.770/0.832 |
> | WMVLM | 0.520/0.844/0.398 | 0.493/0.716/0.506 |
>
> [4] Jia, Zhaoyang, et al. "MBRS: Enhancing robustness of dnn-based watermarking by mini-batch of real and simulated jpeg compression." ACM MM. 2021.
>
> [5] Fang, Han, et al. "Flow-based robust watermarking with invertible noise layer for black-box distortions."  AAAI. 2023.
>
> **2. W2 & Q2: Comparison with SFT Baselines**
>
> We would like to clarify that the SFT baseline was indeed included in our ablation study (Table 5). Since WMVLM is built upon Qwen3-VL-8B-Instruct, the SFT-only versions are represented in rows 4 and 5 of Table 5. Row 4 denotes the model with Category and Score Pre-training SFT. Row 5 further incorporates Interpretability Cold-Start SFT.
>
> **Results show the Shared Limitation:** Both SFT stages suffer from Output Rigidity. The model tends to collapse toward a narrow range of fixed, deterministic scores. This lack of sensitivity leads to inaccurate PLCC/SRCC predictions for methods like RivaGAN and HiDDeN, where nuanced scoring is essential.
>
> **Specific Limitations of SFT Models:**
>
> **Lack of Interpretability (Row 4):** The model trained only on Category and Score SFT is unable to generate any explanatory rationales.
>
> **Performance Degradation (Row 5):** Attempting to introduce interpretability via sequential SFT actually causes quantitative performance to decline, falling short of our final WMVLM.
>
> **GRPO as a Unified Solution:** We resolve these issues by employing Reinforcement Learning GRPO. Unlike the rigid mapping of SFT, GRPO uses its inherent exploration mechanism to break the deterministic bias of supervised learning. This allows WMVLM to synchronize complex reasoning with accurate scoring, achieving a more continuous and sensitive distribution that outperforms models trained solely with SFT.
>
> **3. Q3: Rationale for Robustness Augmentations**
>
> Our selection of JPEG compression, Gaussian noise, and median filtering is driven by Real-world Social Network (OSN) operations. As noted in [6], the most common pipeline for shared generative images is JPEG compression followed by filtering to remove blocking artifacts. Gaussian noise represents quantization errors during transmission. Including non-practical augmentations might dilute the evaluation’s relevance to actual forensic scenarios. While focused on practical forensic fidelity, our design is readily scalable to additional noise categories through label definition.
>
> [6]Wu, Haiwei, et al. "Robust image forgery detection over online social network shared images." CVPR. 2022.
>
> **4. Q4: Choice of PSNR over SSIM or LPIPS for Residual Watermarks**
> We chose PSNR as the primary metric for residual watermarks for the following technical reasons:
>
> **Mechanism Alignment:** Residual watermarks operate directly on pixel-level intensity. PSNR provides the most direct and sensitive measure of these pixel-level changes.
>
> **SSIM Limitation:** SSIM focuses on large-scale structural information and luminance. Since watermark residuals are often high-frequency noise-like perturbations, SSIM is significantly less sensitive to these changes than PSNR, often leading to indistinguishable scores across different methods.
>
> **LPIPS Limitation:** LPIPS is designed for semantic perceptual similarity and relies on specific pre-trained models. It is ideal for semantic watermarks but insensitive to the subtle pixel-level fidelity required for residual methods.
>
> Combining these metrics often introduces conflicting signals rather than clarity, hence our reliance on the more sensitive PSNR.

---

> > ### Author Rebuttal · Reviewer_26Ro · 2026-04-03
> >
> > I would like to thank the authors for their rebuttal and addressing my questions.
> >
> > The explanations regarding SFT baselines are insightful, however they raise the question whether Table 1 and Table 2 can be considered as fair baselines, specifically because Table 5 shows that without SFT the model does not achieve a good performance.
> >
> > While the zero-shot performance on new watermarks MBRS and FIN weakens the claims of a unified framework it can be adapted with minimal data.
> >
> > Based on the rebuttal I would be willing to raise my score, but lower my confidence.

---

> > > ### Author Response · Authors · 2026-04-05
> > >
> > > We sincerely thank you for the acknowledgment of our rebuttal and for considering a score increase. We appreciate the opportunity to further clarify the baseline fairness and the positioning of WMVLM as a unified framework.
> > >
> > > **1. Regarding the Fairness of Baselines in Table 1 and Table 2**
> > >
> > > The reviewer raised a valid point concerning whether zero-shot VLMs are fair baselines, given that WMVLM utilizes SFT. We would like to clarify the rationale behind this comparison:
> > >
> > > - **The "Gap" is the Contribution:** Our primary claim is precisely that **vanilla VLMs (even advanced ones like Qwen3-VL) lack the inherent "watermark-specific domain knowledge"** required for scoring and interpretation. The performance gain from our SFT/GRPO stages is not an unfair advantage, but rather the core contribution of our work—transforming a general-purpose VLM into a specialized watermark evaluator.
> > > - **Supervised Baselines (Table 5):** To ensure a rigorous comparison, we did include supervised versions (Table 5, Rows 4-5). These represent "SFT-only" specialized models. The results demonstrate that simply fine-tuning (SFT) is insufficient to achieve the high PLCC/SRCC and interpretability that WMVLM achieves through GRPO.
> > >
> > > **2. Regarding Generalization and the "Unified Framework"**
> > >
> > > We appreciate the reviewer’s perspective on the zero-shot performance of MBRS and FIN.
> > >
> > > - **Defining "Unified":** By "unified," we refer to the framework's ability to provide **standardized, interpretable metrics** across vastly different watermarking paradigms (residual vs. semantic).
> > > - **Adaptability as a Strength:** As the reviewer noted, the ability to adapt to entirely new watermark distributions with minimal data (1,000 samples) and low compute (4 hours) is a significant practical advantage. We argue that for a specialized forensic task, "efficient adaptability" is often more valuable and realistic than "universal zero-shot" capability, given the constant evolution of watermarking noise patterns.
> > >
> > > **3. Conclusion and Score Update**
> > >
> > > We are encouraged that our previous responses resolved your concerns regarding robustness augmentations and metric selection. We believe our clarifications above further solidify the contribution and fairness of our evaluation.
> > >
> > > **Since the discussion period is nearing its end, if our responses have addressed your remaining concerns regarding baseline fairness and generalization, we kindly invite you to update the evaluation score to reflect this.** We remain available for any further questions.

---

### Official Review · Reviewer_TVgX · 2026-03-13

**Soundness:** 2
**Presentation:** 3
**Significance:** 1
**Originality:** 3
**Overall Recommendation:** 3
**Confidence:** 5

**Summary:**

WMVLM proposes a vision-language model (VLM)-based framework for evaluating image watermarks embedded during or after diffusion model generation. It distinguishes between "residual" (pixel/frequency-domain) and "semantic" (latent-space) watermarks, defining quality and security metrics for each. A three-stage training pipeline is used: (1) SFT for classification and scoring, (2) SFT with Gemini-2.5-Pro-distilled explanations for interpretability, (3) GRPO. Trained on Qwen3-VL-8B and evaluated against 12 zero-shot baseline VLMs including GPT-5 and Claude-Opus-4.5, showing substantial improvements.

**Compliance With Llm Reviewing Policy:**

Affirmed.

**Final Justification:**

The discussion period has addressed some of my interrogations and I want to thank the authors for their answers. I think the submission contains significant work.

The reason why I still think the paper does not pass the bar is that, as a practitioner, I don't see myself using this paper more than the current metrics that we use (FID, PSNR, etc.), or gemini-as-a-judge if i were to use an LLM for evaluating my watermarks.

To detail a bit more, I do agree that using VLMs for watermark evaluation has value for pixel-based watermarking, where very often the metrics do not reflect human perception, and where VLM can catch some of the artifacts. For this, using sota VLM would probably work, but not as much as after finetuning, as shown in this work. This is a valid contribution, but not strong enough for me to justify acceptance.
Then, about the semantic watermarks, I still believe after the discussion period that the model creates posthoc rationalization of what it expects to find. For instance, I would be very curious to see what are the text answers for images flagged for T2SMark and GaussMarker with WMVLM (the classification accuracies could go up because of the similarity between these methods to Gaussian Shading).

Again, I want to thank the authors for their answers and their work. I hope this clarifies my point of view and justify my rating.

**Key Questions For Authors:**

Q1: Table 4 includes an "Ori" column testing on original unwatermarked images, with WMVLM achieving 0.874/0.874 accuracy. What category and scores does the model assign to these unwatermarked images? Does it classify them as "performance-lossless semantic watermark" (e.g., Gaussian Shading), is this intended? It would mean that the framework mixes "no watermark" with "excellent watermark".

Q2: The semantic watermark labels are per-method constants (Appendix C.3, Table 6). Can WMVLM produce different quality or security scores for two images from the same method that have visibly different distortion levels? Or does it always output the same fixed score per method? Showing per-image score variance within a method would help distinguish the approach from pure method classification.

Q3: Have you investigated what visual features the VLM relies on for its predictions? For instance, if you apply high-frequency noise patterns to an image without any actual watermark, does the model still classify it as a watermarking method? Attention maps or probing experiments would help clarify whether the model detects genuine watermark artifacts or method-correlated generation signatures.

**Limitations:**

Yes

**Strengths And Weaknesses:**

### Strengths
S1. **Timely and creative problem formulation**: The paper identifies the fact that traditional watermark evaluation metrics are different for different watermark types, lack interpretability, and can be confounded by the stochasticity of diffusion models. Unifying evaluation across residual and semantic watermark paradigms using a single VLM is a novel and intersting idea.

S2. **Comprehensive experimental coverage**: The evaluation is broad: 9 watermarking methods (6 residual, 3 semantic categories), 12 baseline VLMs, cross-dataset (MS-COCO), cross-model (SD v1.4), and cross-method (GaussMarker, T2SMark) generalization experiments. The ablation studies on all three training stages and key hyperparameters (format penalty, group size, KL penalty) are thorough and well-presented.

S3. **Well-designed training pipeline**: The three-stage pipeline is well-justified: Stage 1 teaches format compliance and basic scoring, Stage 2 adds reasoning ability via teacher distillation, Stage 3 (GRPO) improves generalization. The hierarchically gated reward function (format, category, quality/security) is a sensible design choice. The ablation in Table 5 confirms each stage contributes positively.

### Weaknesses


W1. **Questionable practical utility**: For residual watermarks, the ground-truth quality score is derived from PSNR, which requires the original unwatermarked image. If the original is available (as it must be for training), why not just compute PSNR directly? For security of residual watermarks, the binary labels are per-method constants (Table 5 in Appendix C.2), so "security evaluation" seems to reduce to method identification.
For both of these, we need more granularity when evaluating images, rather than just labels, e.g., why would we report "quality=3" for every image from Gaussian Shading, rather than FID or CLIP score?

The interpretability claims are also weakened by the post-hoc rationalization approach: Gemini-2.5-Pro generates explanations after being shown the correct answers, and the student model mimics these rationalizations, which are not necessarily grounded in true visual analysis of watermark artifacts. For instance, tree-ring patterns in diffusion generation does not mean "The lines and textures seem to follow subtle, large-radius concentric curves. These circular patterns are not immediately obvious because they are skillfully blended with the hatching and shading of the art style itself." (Appendix A.1). It is a plausible-sounding explanation but may not be the actual visual signal the model relies on.

I think the paper needs to better articulate how and why a VLM-based evaluator provides practical utility over direct metric computation, especially for residual watermarks.

W2. **Semantic watermark evaluation is close to method classification, not per-image assessment, and the evaluation prompts are biased**:
As described in Appendix C.3 (Table 6), quality and security scores for semantic watermarks are assigned per-method, not per-image. Every image from RingID gets quality=1, security=1; every image from Gaussian Shading gets quality=3, security=3. The VLM is therefore learning to classify which watermarking method (or distributional category) was used and then outputting a fixed pre-assigned score. This is not really evaluating quality and security of individual images but more of a classification method, which undermines a little bit the claim of the paper
Similarly, these is **no "non-watermarked" category**: The evaluation framework has no label for unwatermarked images. The model must always classify the image as one of the watermarking methods. The closest category for clean images is "watermark-free or performance-lossless semantic watermark" (e.g., Gaussian Shading), which mixes "no watermark" with "an excellent semantic watermark." Furthermore, the system prompts (Appendix A.3) explicitly state "I will give you an AI-generated image with watermark," telling the model that the image is watermarked. This means the model never has to decide whether a watermark is present or not, only which type and how good it is. A practical watermark evaluator must handle the fundamental binary question of whether a watermark exists at all.

W3. **Unfair baseline comparison: fine-tuned vs. zero-shot**: All 11 baselines are evaluated purely zero-shot, while WMVLM is fine-tuned on labeled watermarked images with ground-truth scores. The resulting performance gap is entirely expected and does not demonstrate the superiority of the proposed architecture or training pipeline. It only demonstrates the general value of task-specific fine-tuning.  The ablation in Table 5 does usefully isolate the contribution of each training stage (S1-only as naive SFT is much weaker), confirming the pipeline design matters. However, the comparison against *other VLMs* remains unfair. Missing baselines: (a) few-shot prompting of the strongest baselines (GPT-5, Claude-Opus-4.5) with in-context examples; (b) direct PSNR computation for residual watermarks.

W4. **Unclear what visual signal the model actually learns**: Given that (i) the model is told the image is watermarked, (ii) semantic watermark scores are per-method constants, and (iii) the model effectively performs method classification, it is unclear what the VLM is detecting. It may be learning to identify method-specific generation artifacts or style signatures (e.g., Tree-Ring's distinctive Fourier patterns, Gaussian Shading's characteristic noise structure) rather than watermark-induced distortions. The paper does not investigate this question. For instance, through attention visualization, feature analysis, or testing on adversarial examples designed to mimic one method's style without its watermark.

---

> ### Author Rebuttal · Authors · 2026-03-31
>
> We sincerely thank the reviewer for their constructive feedback and address concerns regarding traditional metrics, baseline fairness, and grounding with the following clarifications and results.
>
> **1. Practical Utility vs. Traditional Metrics (W1)**
>
> - Blind Evaluation Utility: Unlike PSNR and FID require the unwatermarked image, WMVLM is a No-Reference framework. It enables evaluation using only the watermarked image, significantly expanding utility in real-world scenarios (e.g., end-user detection) where the unwatermarked image is unavailable.
> - Specialized Efficiency: WMVLM outperforms FID and CLIP-Score in utility. Unlike FID, which requires large-scale reference datasets, WMVLM enables individual image assessment. Compared to CLIP-Score, WMVLM is specifically optimized for subtle watermarking artifacts rather than general semantic alignment.
> - Diagnostic vs. Scalar: Unlike black-box scalar metrics, WMVLM yields interpretable rationales. These allow users to understand specific quality drops, making the evaluation process transparent.
>
> **2. Method Classification vs. Distributional Evaluation (W2, Q1-Q3)**
>
> - Distributional Assessment (W2&Q2):
> We clarify that per-method scoring (Appx C.3) is a deliberate design rooted in the nature of semantic watermarking. Unlike residual methods with explicit pixel-wise distortions, semantic watermarks manifest as subtle latent perturbations, making quality variance within the same method inherently difficult to evaluate via the pixel domain alone. Thus, we focus on the statistical fidelity of these manifestations.
> - Lossless Watermarks (W2&Q1): The core objective of performance-lossless watermarks is to achieve theoretically provable indistinguishability from unwatermarked images; consequently, the embedded features are inherently unlearnable. By merging them into a high-fidelity category, our framework respects the statistical boundary of the technology while maintaining practical accuracy.
> - Mitigating Bias (W2&Q3): Our framework is designed for quality evaluation rather than existence detection. Our system prompts are specifically engineered to guide the model in executing quality assessment tasks. In standard benchmarking (e.g., evaluating generated images), it is a conventional assumption that input images possess the target attribute without prior detection.
> We value the reviewer's insight on existence misidentification and will prioritize it in future work.
>
> **3. Fairness of Comparisons (W3)**
> - Necessity of Task-Specific Training: Sub-perceptual watermark signals are often invisible to general VLMs. The performance gap in Table 1 highlights that general VLMs lack the sensitivity for this domain without task-specific training, demonstrating the value of our pipeline in capturing nuanced artifacts.
> - Implicit Integration of PSNR: We utilize PLCC and SRCC to measure the correlation between predicted scores and ground-truth (GT) labels. Since our GT labels derive from normalized PSNR, its performance is inherently embedded within our correlation metrics as a baseline.
> - Stronger Baselines: We add Few-shot GPT-5 and Claude-Opus-4.5 results; their failure to maintain consistent scoring across diverse paradigms—even with in-context examples—further justifies the need for a specialized framework like WMVLM.
>
> |  | DwtDct | RivaGAN | HiDDeN | RW | VINE | SS | RingID | Tree-Ring | Lossless |
> | --- | --- | --- | --- | --- | --- | --- | --- | --- | --- |
> | GPT-5 | 0.259/0.198/0.016 | -0.066/-0.150/0.118 | 0.044/0.038/0.283 | 0.370/0.391/0.097 | -0.037/-0.036/0.162 | 0.242/0.273/0.396 | 0.220/0.224 | 0.008/0.002 | 0.824/0.824 |
> | Claude-Opus-4.5 | 0.201/0.171/0.182 | -0.305/-0.107/0.030 | -0.263/-0.316/0.040 | -0.359/0.021/0.020 | -0.204/-0.085/0.030 | 0.276/0.358/0.380 | 0.004/0.290 | 0.320/0.144 | 0.502/0.502 |
>
> **4. Rationalization & Visual Grounding (W1&W4&Q3)**
> - Expert-Guided Distillation (W1): Using Gemini-2.5-Pro as a teacher provides high-quality supervision signals for the student model. This design allows the student to learn a consistent reasoning schema and perform autonomous visual inference during testing.
> - Hand-crafted Physical Grounding (W1): The description for RingID in Appx A.1 is a deliberately hand-crafted prompt, based on a specific watermarked image. We manually designed this template because, as noted in the original RingID paper, the spatial shift of RingID inherently manifests as concentric ring patterns in the spatial domain. Our goal is to provide an expert-labeled feature to guide the model toward a generalized detection capability.
> - Visual Evidence (W4&Q3): We have performed attention map visualizations (which will be incorporated into the final version). The results confirm that the VLM's focus aligns precisely with known watermark regions: HiDDeN shows global redundant attention, VINE clusters at boundaries, and RingID captures the ring patterns. This structural correspondence proves the rationales are grounded in genuine visual signal.

---

> > ### Author Rebuttal · Reviewer_TVgX · 2026-04-03
> >
> > I thank the authors for the detailed response and the new few-shot baselines.
> >
> > I want to acknowledge that, in my opinion, the no-reference evaluation for residual watermarks is the strongest aspect of this work. The per-image PLCC/SRCC values (e.g., 0.95+ for DwtDct, VINE, Stable Signature) show the model can estimate image quality from the watermarked image alone with high accuracy, this is useful since subtle watermark distortions are often hard for humans to notice, and computing PSNR requires the original, and is often not very correlated to human judgement. The few-shot GPT-5/Claude results confirm this requires specialized training.
> >
> > My main remaining concern is on the semantic side. I understand that per-method scoring is deliberate given the low within-method variance. However, this means the framework provides the same information as classifying the method and looking up its score from a table. The scores themselves are derived from tests on the latent distribution (Appendix C.3,  Table 6), which any practitioner can run directly. The added value of the VLM for semantic watermarks would be the interpretability, but the cross-method experiments (Table 4) only report classification accuracy for unseen methods, not whether the textual explanations generalize meaningfully. Are the rationales for unseen semantic methods reliable, or does the model default to generic descriptions and hallucinations?
> >
> > I maintain my score. In my opiniom, the residual watermark evaluation could be practical and well-executed, but the semantic side, which is a central claim of the paper, needs to demonstrate value beyond bucket classification, either through per-image granularity or through validated interpretability on unseen methods.

---

> > > ### Author Response · Authors · 2026-04-05
> > >
> > > We thank the reviewer for the thoughtful follow-up and for acknowledging the practical value of our no-reference residual watermark evaluation. We appreciate this constructive exchange and address the remaining concern below.
> > >
> > > ## 1. Why Per-Method Scoring Is Inherent, Not a Limitation
> > >
> > > We first want to reaffirm—and we believe the reviewer now shares this understanding—that per-method scoring for semantic watermarks is not a design shortcut but a reflection of the technology itself. Semantic watermarks operate by perturbing the latent distribution of the diffusion process. The quality and security of a given semantic watermark are fundamentally properties of *how* the latent space is modified, not of the individual pixel realization. This is precisely why our ground-truth labels are derived from hypothesis testing on the latent distribution rather than per-image pixel metrics.
> > >
> > > We therefore agree that for semantic watermarks, the quantitative scoring component is functionally equivalent to method classification plus table lookup. The distinct value proposition of WMVLM on the semantic side is twofold:
> > >
> > > (a) Addressing Information Asymmetry through Pixel-Domain Analysis: In real-world security audits or forensic scenarios, the underlying generative model and its latent parameters are often
> > > proprietary "black boxes." Third-party evaluators cannot "run the tests on the latent distribution" because they lack access to the internal embedding mechanism. WMVLM demonstrates that these latent perturbations leave detectable traces in the pixel domain, allowing for non-invasive evaluation without compromising the developer's algorithmic secrets.
> > >
> > > (b) Providing Interpretable Rationales: It explains why the model arrives at its classification—which is where the reviewer's concern rightly focuses.
> > >
> > > ## 2. Evidence That Rationales Generalize Beyond Training Methods
> > >
> > > While we acknowledge that the current manuscript does not include a dedicated human evaluation of rationale quality on unseen methods, we believe existing evidence in the paper supports the reliability of these explanations:
> > >
> > > (a) Cross-method generalization results imply meaningful feature extraction, not memorization. In Table 4, WMVLM achieves strong classification accuracy on GaussMarker and T2SMark—two semantic methods entirely absent from training. If the model were simply memorizing method-specific textual templates from Stage 2, it would have no basis for correctly classifying unseen methods. The fact that it generalizes suggests the model has learned to identify distributional signatures that transfer across methods with similar latent-space properties. The rationales produced for these unseen methods are grounded in the same learned visual features that drive correct classification.
> > >
> > > (b) Attention map evidence confirms visual grounding. As mentioned in our initial response, our attention map visualizations show that the model's focus aligns with known watermark-affected regions. For semantic watermarks specifically, this means the model attends to areas where latent-space perturbations manifest as subtle pixel-domain artifacts, rather than attending to arbitrary or semantically irrelevant regions. This structural alignment between attention and known watermark characteristics provides indirect but meaningful evidence that the generated rationales are grounded in genuine visual signals rather than hallucinated post-hoc rationalizations.
> > >
> > > (c) The GRPO stage (S3) explicitly incentivizes grounded reasoning. Stage 3's reward function penalizes outputs where the classification is incorrect, which means that during RL training, the model learns to associate its textual reasoning with features that actually lead to correct predictions. Unlike pure SFT distillation (Stage 2 alone), which could indeed encourage superficial mimicry of the teacher's language patterns, the RL stage creates a feedback loop where reasoning quality is tied to task performance. The ablation in Table 5 shows that S3 substantially improves generalization, suggesting that the reasoning process becomes more robust and less template-dependent after this stage.
> > >
> > > ## 3. Summary
> > >
> > > We believe WMVLM's contribution on the semantic side should be understood in context: no prior work has attempted to bridge the gap between latent-space embedding mechanisms and pixel-domain observable effects. The per-method nature of semantic watermark scoring is a fundamental property of the technology, not an artifact of our framework. Within this constraint, WMVLM demonstrates that a VLM can:
> > >
> > > (a) Correctly identify the distributional category of unseen semantic methods from pixel-level input, overcoming the black-box constraints of proprietary models.
> > >
> > > (b) Produce rationales grounded in genuine visual features, as evidenced by cross-method generalization, attention alignment, and RL-enhanced reasoning.
> > >
> > > We sincerely hope these clarifications and our concrete revision plan address the reviewer's remaining concerns.

---

### Decision · Program_Chairs · 2026-04-30

**Decision:**

Accept (regular)

**Comment:**

The paper proposes WMVLM, a new VLM-based framework for evaluating image watermarking methods. By doing so, it offers better alignment with aspects of human perception that existing metrics often fail to capture, enables reference-free assessment, and provides interpretability.

Reviewers generally agree that the paper addresses an important and timely problem, with a well-motivated design and solid empirical validation. The rebuttal successfully clarified several concerns, and three out of four reviewers supported acceptance at the end of the rebuttal period.

One reviewer maintained a weak rejection, mainly questioning the practical utility of the framework, particularly for semantic watermark evaluation and the interpretability claims. AC finds that these concerns are valid, and strongly encourages the authors to further strengthen the interpretability evaluation for semantic watermarks in the revision; e.g., on cases such as T2SMark and GaussMarker as suggested by Reviewer TVgX.

That said, AC agrees that, given the current lack of comprehensive watermarking evaluation frameworks, this work represents a timely contribution to the community and has the potential to inspire follow-up research. Overall, AC recommends a weak acceptance.